# Ssd1 and Gcn2 suppress global translation efficiency in replicatively aged yeast while their activation extends lifespan

**Zheng Hu[1][†], Bo Xia[2,3][†], Spike DL Postnikoff[1], Zih-Jie Shen[1], Alin S Tomoiaga[2,3,4], Troy A Harkness[5], Ja Hwan Seol[6], Wei Li[7], Kaifu Chen[2,3]\*, Jessica K Tyler[1]\***

[1]Department of Pathology and Laboratory Medicine, Weill Cornell Medicine, New York, United States; [2]Department of Cardiovascular Sciences, Houston Methodist Research Institute, Houston, United States; [3]Department of Cardiothoracic Surgery, Weill Cornell Medicine, New York, United States; [4]Manhattan College, Bronx, United States; [5]Department of Anatomy and Cell Biology, University of Saskatchewan, Saskatoon, Canada; [6]Department of Epigenetics and Molecular Carcinogenesis, University of Texas MD Anderson Cancer Center, Houston, United States; [7]Dan L. Duncan Cancer Center and Department of Molecular and Cellular Biology, Baylor College of Medicine, Houston, United States

**\*For correspondence:**
kchen2@houstonmethodist.org (KC);
jet2021@med.cornell.edu (JKT)

[†]These authors contributed equally to this work

**Abstract** Translational efficiency correlates with longevity, yet its role in lifespan determination remains unclear. Using ribosome profiling, translation efficiency is globally reduced during replicative aging in budding yeast by at least two mechanisms: Firstly, Ssd1 is induced during aging, sequestering mRNAs to P-bodies. Furthermore, Ssd1 overexpression in young cells reduced translation and extended lifespan, while loss of Ssd1 reduced the translational deficit of old cells and shortened lifespan. Secondly, phosphorylation of eIF2$\alpha$, mediated by the stress kinase Gcn2, was elevated in old cells, contributing to the global reduction in translation without detectable induction of the downstream Gcn4 transcriptional activator. tRNA overexpression activated Gcn2 in young cells and extended lifespan in a manner dependent on Gcn4. Moreover, overexpression of Gcn4 sufficed to extend lifespan in an autophagy-dependent manner in the absence of changes in global translation, indicating that Gcn4-mediated autophagy induction is the ultimate downstream target of activated Gcn2, to extend lifespan.
DOI: https://doi.org/10.7554/eLife.35551.001

## Introduction

Aging is a complex biological process shared by all living organisms. However the molecular causes of aging are poorly understood. While aging studies in mammals are slow due to their long life-spans, the single-celled eukaryote budding yeast has become a leading model system for aging studies due to its short lifespan, ease of genetic and environmental manipulations, and conserved aging pathways (*Bitterman et al., 2003*; *Steinkraus et al., 2008*). Furthermore, budding yeast represent the only system in which the number of times a cell divides, also known as replicative lifespan, can be accurately determined. Together with recently developed methodologies to isolate larger quantities of old yeast cells, such as the mother enrichment program (MEP) (*Lindstrom and Gottsch-ling, 2009*), identification and characterization of the molecular changes that accompany, and potentially cause, replicative aging is now feasible, enabling many questions about aging to be answered.

For example, how does protein synthesis change during replicative aging and do these changes promote a maximal longevity benefit?

Protein synthesis is an essential cellular process affecting growth, reproduction and survival, and whose activity changes in response to both intrinsic and extrinsic cues such as nutrient availability and energy levels. When nutrients are abundant and cells are growing, the target of rapamycin (TOR) kinase is active, phosphorylating and inactivating the 4EBP1 repressor of CAP-dependent translation and activating the S6 kinase 1 to promote ribosome biogenesis via phosphorylation of Rps6 (*Blenis, 2017*). In response to limited growth or limited nutrients, global CAP-dependent translation is repressed due to TOR inactivation and the accompanying dephosphorylation of 4EBP1 and Sch9 in mammalian cells, while in yeast TOR inactivation reduces global translation via reduced gene expression of ribosomal protein genes (*Martin et al., 2004*). A distinct parallel pathway that reduces global protein synthesis is stress-induced activation of the stress response kinase(s) that result in inhibitory phosphorylation of eIF2α (*Donnelly et al., 2013*). There are multiple stress response kinases that phosphorylate eIF2α in mammals in response to different stresses, but in yeast the only eIF2α kinase is Gcn2. Yeast Gcn2 is activated by salt stress, acid, oxidative stress and by the uncharged tRNAs that accumulate when intracellular amino acid levels are low (*Gallinetti et al., 2013*). Phosphorylation of eIF2α results in globally reduced efficiency of translational initiation while paradoxically inducing translation of the Gcn4 transcriptional regulator (the yeast counterpart of ATF4) due to leaky scanning of ribosomes through its inhibitory upstream open reading frames (uORFs) to the *GCN4* ORF (*Gallinetti et al., 2013*). Gcn4 induces expression of a variety of genes that mediate amino acid biosynthesis, purine biosynthesis, organelle biosynthesis, ER stress response, mitochondrial carrier proteins and autophagy (*Pakos-Zebrucka et al., 2016*), while also repressing genes encoding the translation machinery and ribosomes (*Mittal et al., 2017*). As such, cells respond to many forms of stress by down-regulation of protein synthesis at both the translational initiation stage and transcriptional repression of the translation machinery.

Manipulations that mildly lower the rate of protein synthesis often also lower the rate of aging, increasing the lifespan of organisms from yeast to humans (*Tavernarakis, 2008*). For example, the TOR pathway is a conserved player in longevity, where it regulates many processes such as transcription, autophagy, cytoskeletal organization, protein turnover and mRNA translation (*Laplante and Sabatini, 2012*). Inactivation of TOR, for example by the drug rapamycin, reduces protein synthesis and extends lifespan in organisms from yeast to mice (*Blagosklonny, 2013*). However, given that TOR affects multiple physiological processes, it is unclear how much of the lifespan-extending benefit of TOR inhibition is via its role in controlling protein synthesis. More direct evidence supporting a role for mildly reduced protein synthesis in increasing organismal longevity comes from knockdown or deletion of genes encoding the translational machinery itself. The rate of translational initiation is largely controlled by eukaryotic translation initiation factors (eIFs). In particular, eIF4E facilitates the recruitment of ribosomes to the mRNA, which is a major rate-limiting step in protein synthesis. Loss of one specific isoform of eIF4E in *C. elegans* extends lifespan (*Syntichaki et al., 2007*). Similarly, reducing the levels of other eIFs, or certain large ribosomal subunits, reduces protein synthesis and extends organismal lifespan in worms, flies and yeast (*Hansen et al., 2007*; *Pan et al., 2007*; *Chen et al., 2007*; *Curran and Ruvkun, 2007*; *Steffen et al., 2008*; *McCormick et al., 2015*). The protein synthesis inhibitor cycloheximide also extends lifespan in *C. elegans* and delays senescence in normal human fibroblasts (*Takauji et al., 2016*). However, not all manipulations that reduce global protein synthesis extend lifespan, such as depletion of most yeast small ribosomal subunits (*Steffen et al., 2008*). Moreover, for the manipulations that reduce general protein synthesis and increase lifespan, it is not clear whether the reduced protein synthesis per se causes lifespan extension or just correlates with it. Notably, the full yeast lifespan extension that results from depletion of large ribosomal subunits, *TOR1* deletion, or dietary restriction, requires the transcriptional regulator Gcn4 (*Steffen et al., 2008*). Which of the many processes transcriptionally controlled by Gcn4, that is key for lifespan extension, is currently unknown.

During the normal aging process, where examined, global protein synthesis generally declines with increased organismal age (*Tavernarakis, 2008*). Conversely, elevated levels of protein synthesis have been observed during premature aging, as seen in Hutchinson-Gilford progeria syndrome (*Buchwalter and Hetzer, 2017*). Analyses of protein synthesis during aging to date have examined bulk protein synthesis not the translation of specific transcripts, so we don't really know which proteins are being most affected. Furthermore, the molecular cause of reduced protein synthesis during

aging is unknown. More specifically, whether protein synthesis is reduced during replicative aging, as opposed to organismal aging, has not been examined in any type of eukaryotic cell.

We previously suggested that histone protein synthesis may be reduced during yeast replicative aging (*Feser et al., 2010*). This was based on the observation that levels of histone proteins go down during replicative aging, causing aging, despite increased histone transcript levels and no change in the half-life of histone proteins. To investigate directly whether protein synthesis is altered in old cells, we performed the first genome-wide analysis of protein synthesis during replicative aging. Using ribosome profiling, we found that the translational efficiency of most transcripts is reduced during replicative aging, while translational efficiency did not increase significantly for any mRNA transcript. Mechanistically, elevated levels of the mRNA binding protein Ssd1 in old cells reduced protein synthesis, by delivering mRNAs to aggregated P-bodies, which we found to be at drastically increased levels in old cells. Furthermore, the Ssd1-mediated reduction in protein synthesis during aging enabled optimization of the normal lifespan, because Ssd1 loss shortened lifespan while Ssd1 overexpression extended lifespan. In parallel, the phosphorylation of the Gcn2 substrate eIF2α is increased in old cells, further reducing global protein synthesis. Activation of Gcn2 in young cells by expression of uncharged tRNAs, or overexpression of its downstream effector Gcn4 extended lifespan in a manner dependent on autophagy.

## Results

### Protein synthesis is globally down-regulated in yeast during replicative aging

We have shown previously that the abundance of virtually every transcript encoded by the yeast genome increases during replicative aging, with the exception of those that are highly abundant already, such as those encoding the ribosome protein genes, which do not significantly change during aging (*Hu et al., 2014*). To determine whether increased transcript abundance leads to more protein synthesis, we examined global protein synthesis levels during aging. In these analyses, we isolated budding yeast cells that had undergone an average of 25–30 divisions, where the median replicative lifespan is 22 divisions, using the MEP as described previously (*Hu et al., 2014*). Importantly, this is much older than most researchers can achieve and the MEP, coupled with biotin affinity purification of old cells, enables isolation of much greater quantities of old cells than other approaches. We performed polysome profiling to compare how much total RNA was engaged with ribosomes in old versus young cells. During protein synthesis, the relative monosome versus polysome occupancy in a polysome profiling analysis is a function of initiation versus total elongation time. That is, if less time is spent during initiation than elongation, mRNA will be predominantly polysome-associated. Conversely, if the initiation is much slower than elongation, RNA will be predominantly monosome-associated. A high initiation:elongation ratio can be driven either by ORF length or by slow initiation rate, often indicative of translation regulation. While young cells had mRNAs bound to 1–5 ribosomes, old cells showed extremely low levels of RNA on polysomes (*Figure 1A*). In old cells, the RNA was not only less polysome-associated but also less monosome-associated, providing evidence that translational initiation is extremely low (*Figure 1A*). This is not a consequence of accumulation of dead cells, because the old cells isolated by the mother enrichment program are still viable (*Hu et al., 2014*). Notably, rRNA levels are equivalent during aging (*Pal et al., 2018*). These results indicate that there is a striking defect in global protein synthesis in replicatively aged yeast.

### The mRNA binding protein Ssd1 and aggregated P-bodies are induced during yeast replicative aging

In an effort to uncover genes that influence longevity, we identified a small subset of genes that were upregulated during aging (which was mostly due to histone loss during aging), but not upon experimental depletion of histone H3 (*Hu et al., 2014*). One of these genes, *SSD1*, was of particular interest with regard to reduced protein synthesis during aging, because it is an mRNA binding protein that can function as a translational repressor (*Jansen et al., 2009*). Furthermore, a C-terminal truncation mutation within the *SSD1* gene had been shown to be responsible for the short lifespan of certain yeast strain backgrounds, while long-lived yeast strain backgrounds have wild type *SSD1*

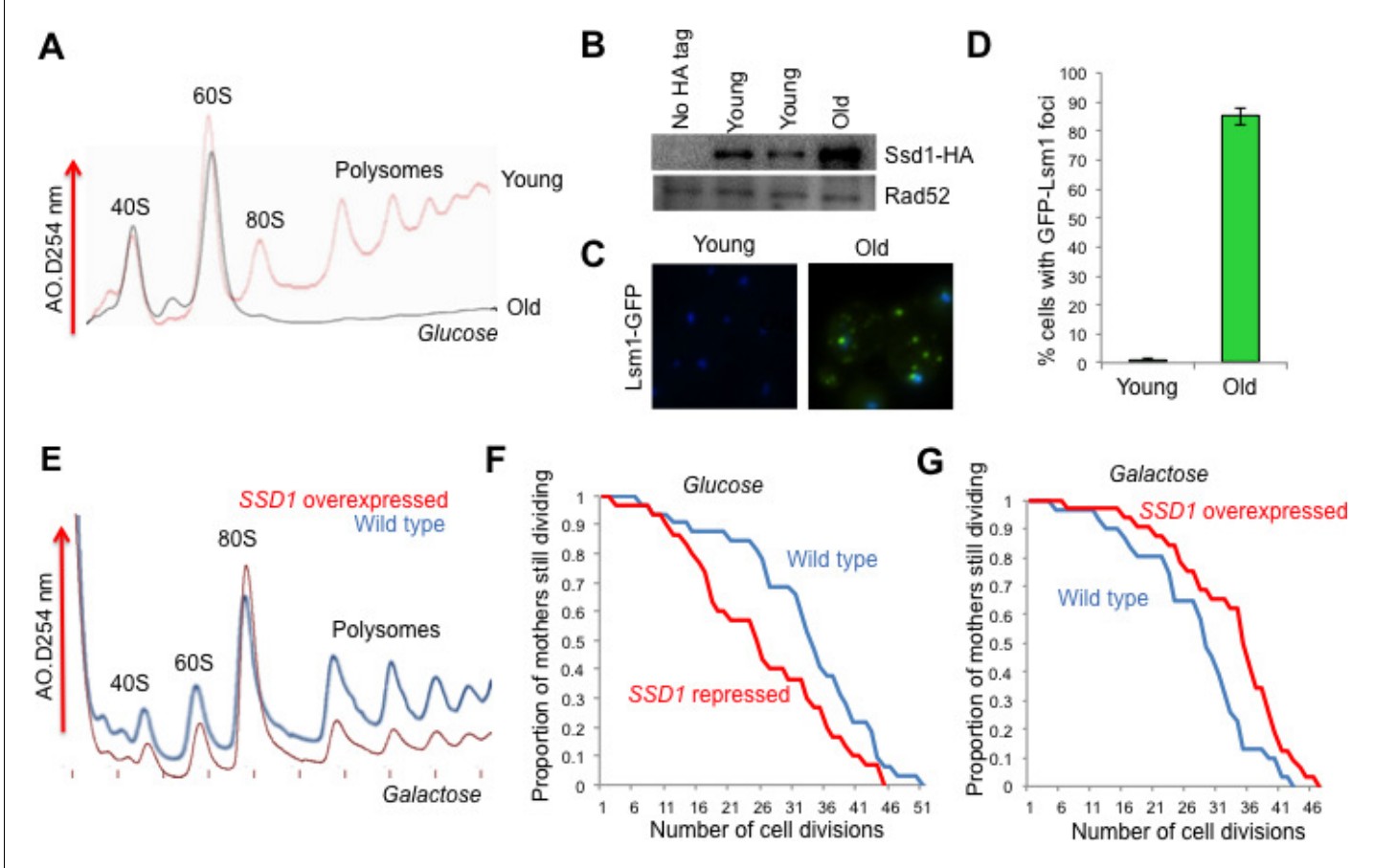

**Figure 1.** The role of Ssd1 in translational regulation and replicative lifespan. (**A**). Polysome analysis of yeast cell lysates. UV traces of RNA distribution on sucrose gradients from young and old ZHY2 cells. (**B**). Western blot analysis of HA tagged Ssd1 protein where endogenous Ssd1 was tagged with HA at the C-terminus (ZHY6), from an equal number of young or old yeast cells. Two independent young samples were analyzed. Cells with no HA tag serve as a control (strain ZHY2). Rad52 serves as a loading control. (**C**). The number of cells showing visible P-body aggregation is increased during aging. WT cells expressing a GFP tagged version of Lsm1 (strain ZHY9) are shown in young and old cells. Young and old cells were stained with DAPI. (**D**). Quantitation of the percentage of young and old cells with Lsm1-GFP foci. For each cell type, an average of 100 cells were examined in three independent experiments (n = 3, mean ±SD). (**E**) Polysome analysis of yeast cell lysates. UV traces of RNA distribution on sucrose gradients from WT cells (strain ZYH2) and *ssd1Δ* cells overexpressing Ssd1 under the *GAL1* promoter (strain ZYH3) grown in 0.5% galactose. (**F**) Replicative lifespan of yeast strain ZHY2 (wild type) and the isogenic strain expressing pGAL-SSD1 (strain ZHY3) grown on glucose to repress the pGAL promoter. (**G**) the same strains as F. grown on 0.5% galactose medium to induce the expression of Ssd1 from pGAL-*SSD1*.

DOI: https://doi.org/10.7554/eLife.35551.002

(*Kaeberlein et al., 2004*). Given the profound defect in protein synthesis during replicative aging (*Figure 1A*), we could not assume that elevated transcript levels of *SSD1* would equate to more Ssd1 protein in old cells. Therefore, we confirmed that Ssd1 protein levels were in fact greatly increased in old cells (*Figure 1B*). Ssd1 normally delivers mRNAs to daughter cells for translation but in response to stress, Ssd1 transports mRNAs to cytoplasmic aggregates of P-bodies (*Kurischko et al., 2011*), Given the increase in Ssd1 during aging (*Figure 1B*), we asked whether aggregated P-bodies visibly accumulate during aging. Aggregated P-bodies, visualized as cytoplasmic puncta containing the P-body protein Lsm1 (Lsm1-GFP) and Pab1-GFP (data not shown), became much more abundant in old yeast (*Figure 1C*), where more than 85% of old cells showed visible P-bodies compared to only 1% in young cells (*Figure 1D*). In addition, Ssd1-GFP co-localized with P-bodies in old cells (data not shown). These results show that aggregated P-bodies accumulate during replicative aging and it is likely that the age-induced Ssd1 targets mRNA to the P-bodies leading to down-regulation of their protein synthesis in old cells.

## Overexpression of Ssd1 in young cells represses protein synthesis and extends lifespan

To investigate whether elevated levels of Ssd1 in old cells has the potential to reduce global protein synthesis, we examined the consequences of overexpressing Ssd1 in young cells. High-level overexpression of Ssd1 is known to be lethal (*Sopko et al., 2006*), so we moderately overexpressed Ssd1 using 0.5% galactose to induce *SSD1* under the galactose-inducible *GAL1* promoter. Cell growth is slower in galactose, accounting for the reduced protein synthesis in the wild type cells apparent in the polysome profile (*Figure 1E*) as compared to yeast grown in glucose (*Figure 1A*). Despite this, it is clear that moderate overexpression of Ssd1 reduced protein synthesis compared to that in wild type cells (*Figure 1E*). In a growing culture of yeast, 50% of the cells have divided 0 times, 25% have divided once, 12.5% have divided twice etc., such that most cells in a yeast culture are predominantly young. As such, these data indicate that elevated Ssd1 levels are sufficient to reduce protein synthesis in young cells.

Given that reduced protein synthesis generally correlates with extended lifespan, we asked whether moderate overexpression of Ssd1 extends lifespan and conversely whether repression of Ssd1 shortens lifespan. The absence of Ssd1, achieved by glucose–mediated repression of p*GAL1-SSD1*, significantly shortened replicative lifespan (*Figure 1F*). Conversely, moderate overexpression of Ssd1 significantly extended lifespan (*Figure 1G*). These results indicate that Ssd1 function, presumably in translational repression, is required for yeast to achieve a normal lifespan, while overexpression of Ssd1 can extend yeast lifespan. Taken together with the reduced protein synthesis that occurs upon Ssd1 overexpression in young cells (*Figure 1E*), these data suggest that the age-induced increase in Ssd1 levels may be at least partially responsible for the age-related decline of global protein synthesis.

## Ribo-seq analysis of protein translation during replicative aging

To understand the nature of the defect in protein synthesis during replicative aging (*Figure 1A*), and to identify which mRNAs undergo reduced translation, we performed ribosome profiling, also called Ribo-seq (*Ingolia, 2014*; *Ingolia, 2016*). In order to detect global changes in protein synthesis between young and old cells, we included spike-in controls to enable normalization back to the number of cells per sample (*Chen et al., 2015*). Ribo-seq analysis, performed three times from independent samples for young, middle aged and old yeast, showed that translation in general decreases during aging, as seen by the increased amount of blue and reduction of yellow in the Ribo-seq heat map, as the cells became older (*Figure 2A*). Given that all transcripts become more abundant during aging (*Hu et al., 2014*), we needed to determine the translational efficiency for each transcript during aging. Therefore, we performed RNA-seq analysis in parallel on the same samples used for the RIbo-seq analyses. This confirmed that the abundance of m transcripts increase during aging (*Figure 2B*) (*Hu et al., 2014*). We also observed an increase in cryptic (*Figure 2—figure supplement 1*) transcription within genes during aging, as seen previously (*Sen et al., 2015*) and intergenic (*Figure 2—figure supplement 2*) transcription during aging consistent with the reduced histone occupancy during aging (*Hu et al., 2014*) promoting access by the general transcription machinery to cryptic initiation sites. To obtain the translational efficiency for each transcript, we normalized the Ribo-seq data to the RNA-seq data, using the data analysis steps shown in (*Figure 2—figure supplement 3*). This dataset represents the first detailed examination of protein synthesis during replicative aging in any organism (*Figure 2C,D*, *Supplementary file 2*). A representative screen shot of the Ribo-seq analysis during aging is shown in (*Figure 2—figure supplement 4*).

Strikingly, translational efficiency decreased globally during aging, apparent from the increased amount of blue color in the translational efficiency heat map, as the cells got older (*Figure 2C*). Notably, all the data shown in *Figure 2A–D* are ranked in the identical manner, with genes at the top being those whose translational efficiency was most reduced during aging and those at the bottom being genes with the least change in translational efficiency during aging. Translational efficiency significantly decreased up to 40 fold during aging for 59% of the genes in the yeast genome (3472 out of 6553), while the remainder of the transcripts had no significant change in translational efficiency during aging (*Figure 2C,D*). The reduction in translational efficiency was already apparent in middle-aged yeast (median age of 12 divisions), albeit becoming more pronounced in the old yeast (*Figure 2C*) suggesting that the translational reduction is not merely reflecting the status of

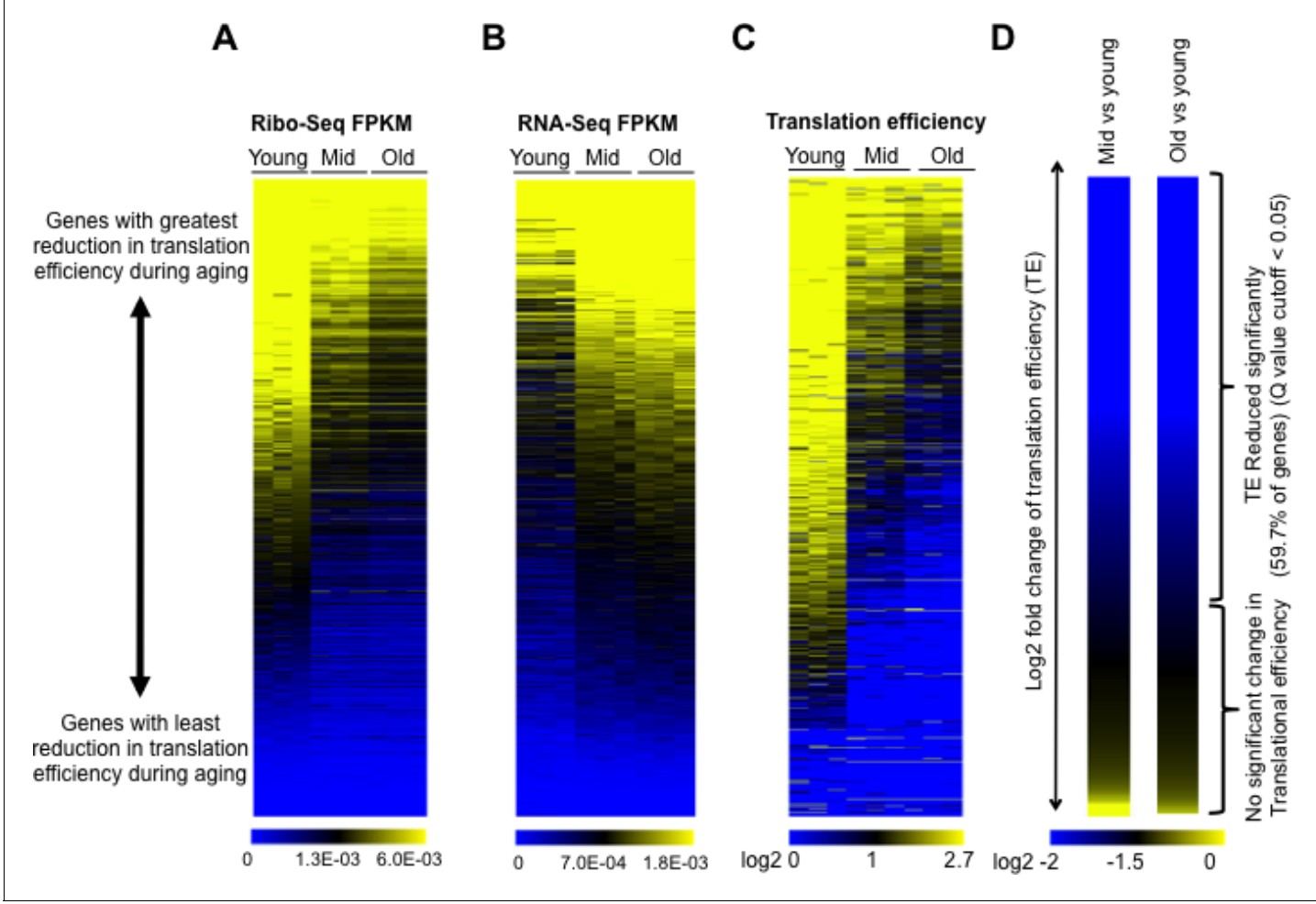

**Figure 2.** Global reduction in protein synthesis during aging. (A) Ribo-seq, (B) RNA-seq, (C) Translational efficiency, and (D) efficiency change of every mRNA obtained by normalizing the Ribo-seq data to RNA-seq data, after normalizing both data sets to spike-in controls. All panels had genes ranked as shown in D, by the Log2 fold change of translation efficiency between old and young cells, with the genes with the biggest reduction in translation efficiency on the top and the genes with the least changes in translational efficiency at the bottom. Data for three independent experimental repeats is shown (Q value cutoff < 0.05) for young, middle aged (mid) and old cells. All data was obtained using strain ZYH2. See also *Figure 2—figure supplements 1–5*.

DOI: https://doi.org/10.7554/eLife.35551.003

The following figure supplements are available for figure 2:

**Figure supplement 1.** Internal transcription initiation increases with age.

DOI: https://doi.org/10.7554/eLife.35551.004

**Figure supplement 2.** Intergenic transcription initiation increases with age.

DOI: https://doi.org/10.7554/eLife.35551.005

**Figure supplement 3.** Overview of steps used in the analysis of the Ribo-seq Data.

DOI: https://doi.org/10.7554/eLife.35551.006

**Figure supplement 4.** Screen shot of typical genomic region, showing Ribo-seq and RNA-seq.

DOI: https://doi.org/10.7554/eLife.35551.007

**Figure supplement 5.** translational initiation not elongation is reduced during replicative aging.

DOI: https://doi.org/10.7554/eLife.35551.008

the cells within the population that had completed their cell divisions. We found no evidence for increased translational pausing or elongation defects during aging which are characterized by increased peaks of ribosomes within ORFs and general enrichment of ribosomes within the ORF relative to the start of the ORF (*Ingolia, 2016*), respectively (*Figure 2—figure supplement 5*) suggesting that the rate-limiting step in protein synthesis during aging is initiation.

Upon comparing the transcript abundance for every gene in young cells to the change in translational efficiency during aging, it was apparent that translation efficiency went down the most during aging for the transcripts that were the most abundant in young cells (*Figure 3A*). The greatest reduction in translational efficiency during aging was seen for genes encoding ribosomal proteins, or other proteins required for protein synthesis or its regulation (*Figure 3B–D*, *Figure 3—figure supplements 1* and *2*). Transcripts that were least abundant in young cells did not have a significant change in translation efficiency (*Figure 3A*) and functioned in a broad variety of pathways (*Figure 3—figure supplement 3*).

## Increased phosphorylation of the Gcn2 kinase target eIF2α and persistent phosphorylation of the TOR kinase target Rps6 during replicative aging

Given the profound decrease in protein synthesis during aging (*Figure 2*), while Ssd1 has been reported to associate with only 59–152 specific mRNAs (*Jansen et al., 2009*; *Hogan et al., 2008*), we considered that additional mechanisms must contribute to the decline in global protein synthesis during aging. Because activation of Gcn2 and inhibition of TOR globally decreases protein synthesis, we examined whether either of these events occurred in old cells. We found that phosphorylation of the Gcn2 target eIF2α is clearly apparent in old but not young cells (*Figure 4A*). The level of eIF2α phosphorylation in old cells was almost as much as that achieved by depletion of amino acids to 20% of their normal levels, which is a known condition that activates Gcn2 (*Figure 4B*). By contrast, striking TOR inhibition was not apparent in old cells, because the S6K1-mediated phosphorylation of Rps6 persisted (*Figure 4B*). These data are consistent with old cells experiencing a stress condition that is sensed specifically by Gcn2.

To examine the similarities between the translational changes that result from Gcn2 activation and those that occur during aging, we compared our replicative aging Ribo-seq data with the Ribo-seq data obtained during amino acid depletion (*Ingolia et al., 2009*). Although the analysis from the Weissman lab only found 111 genes with significantly reduced translational efficiency during amino acid depletion, as compared to the 3472 with significantly reduced translational efficiency during aging, this likely reflects that they did not include spike-in controls to detect global changes in translational efficiency. However, even without spike-in controls, the ranking of the genes and their relative change in translational efficiency is still valid, even though the absolute change in translational efficiency and direction of increase vs. decrease is inaccurate. Accordingly, we observed a positive correlation (r = 0.614) between the two data sets, where the genes with the greatest reduction in translational efficiency during replicative aging tended to have the greatest reduction in translational efficiency during Gcn2 activation by amino acid starvation (*Figure 4C*, *Figure 4—figure supplement 1*). In both cases, the genes with the greatest reduction in translational efficiency encoded proteins involved in ribosome biogenesis and translational regulation (*Ingolia et al., 2009*) (*Figure 3—figure supplement 1*, *Figure 4—figure supplement 1*). Many of the genes with the least change in translational efficiency during aging were at too low abundance to be detected in the Weissman lab's Ribo-seq data set (*Figure 4—figure supplement 2*), and therefore were left out of the comparison (*Figure 4C*). As such, the decreases in translational efficiency that occur during aging mimic the decreases in translational efficiency that occur upon Gcn2 activation by amino acid depletion.

We found that 122 transcripts have increased translation during amino acid depletion but not during aging (*Figure 4C*), and these tend to encode proteins involved in the stress response pathways that are induced by amino acid starvation (*Ingolia et al., 2009*) (*Figure 4D*). The genes that had increased translation during amino acid depletion but not during aging were significantly enriched in the 40 genes that are upregulated during Gcn4 overexpression and with the approximately 60 genes with Gcn4 consensus sites in their promoters (*Figure 4—figure supplement 3*) (*Mittal et al., 2017*). In agreement with failure to observe translation of the transcripts activated by Gcn4 during aging, we find that even though levels of phosphorylated eIF2α are higher during aging (*Figure 4A,B*), we do not find significant upregulation of a Gcn4-luciferase reporter in old cells (*Figure 4E*). Furthermore, translation of the endogenous Gcn4 was not increased in old cells, apparent in the ribosome profile over the Gcn4 transcript (*Figure 4F*). While there was less translation of the *GCN4* upstream ORFs in old cells, this was not sufficient to lead to increased translation of the Gcn4 ORF (*Figure 4F*). Consistent with this, we did not observe preferential transcriptional activation or repression of Gcn4 targets during aging (*Hu et al., 2014*). In summary, consistent with the elevated levels

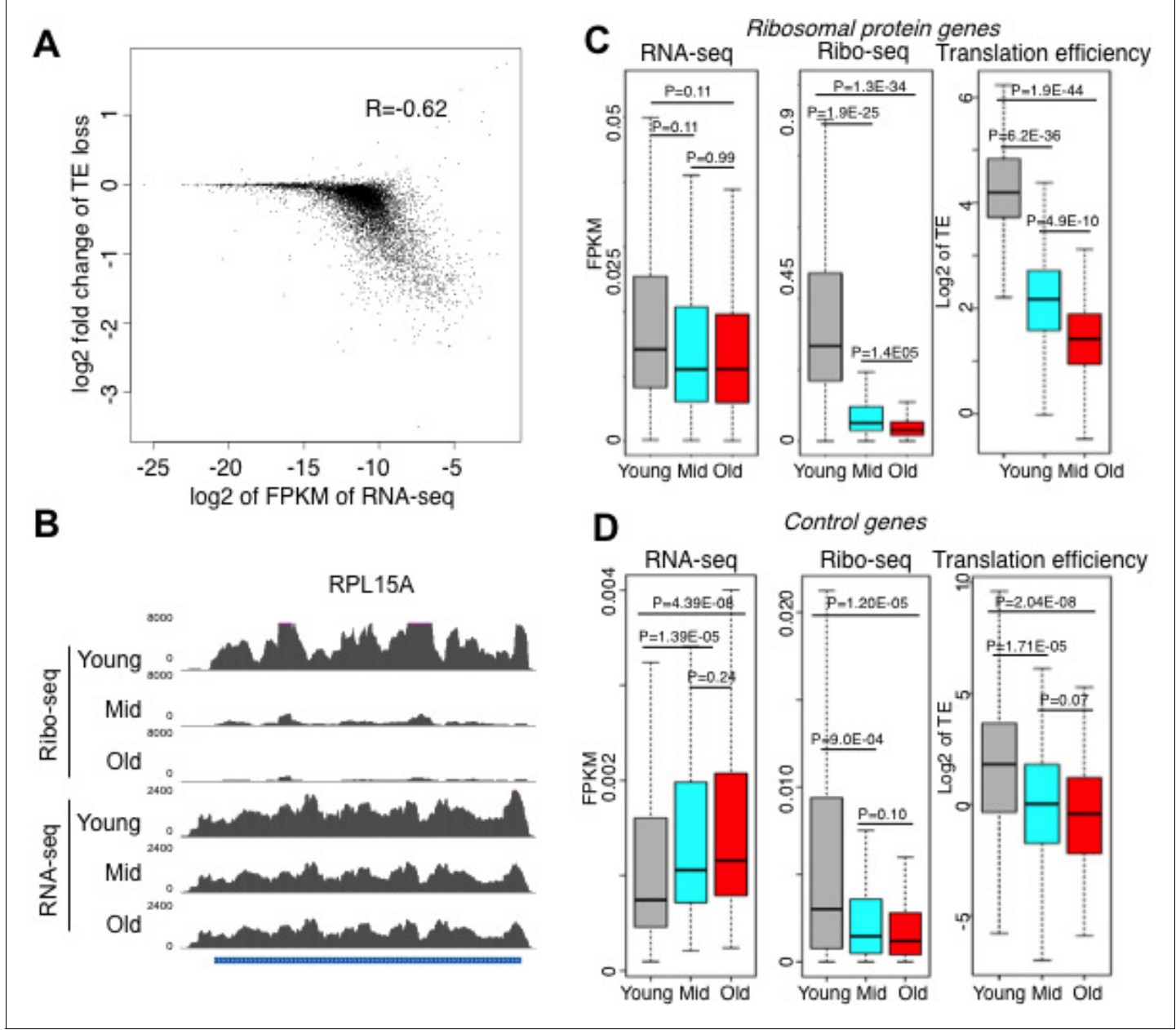

**Figure 3.** Reduced initiation of protein synthesis during aging; genes that are most highly transcribed in young cells having the greatest reduction in protein synthesis. (A) Scatter plot of log2 fold change of translational efficiency reduction during aging and log2 of FPKM (RNA abundance) in young cells. (B). Screen shot of ribosomal protein gene RPL15A, showing the typical reduction in translation efficiency, with no significant change in transcript levels during aging. (C). RNA-seq and Ribosome profiling and translation efficiency of the ribosome protein genes (RPG) during aging. P values are calculated by Wilcoxon test. While the reduction in transcript levels for all 136 of the RPGs is significant, each RPG transcript decreases on average only 23% during aging. (D). As for C, but for 100 randomly chosen genes. See also *Figure 3—figure supplements 1–3*.

DOI: https://doi.org/10.7554/eLife.35551.009

The following figure supplements are available for figure 3:

**Figure supplement 1.** GO term analysis for genes whose translation efficiency went down the most during aging.

DOI: https://doi.org/10.7554/eLife.35551.010

**Figure supplement 2.** Translation efficiency (TE) is reduced the most during aging for genes that were highly expressed in young cells.

DOI: https://doi.org/10.7554/eLife.35551.011

**Figure supplement 3.** GO term analysis for genes whose translation efficiency went down the least during aging.

DOI: https://doi.org/10.7554/eLife.35551.012

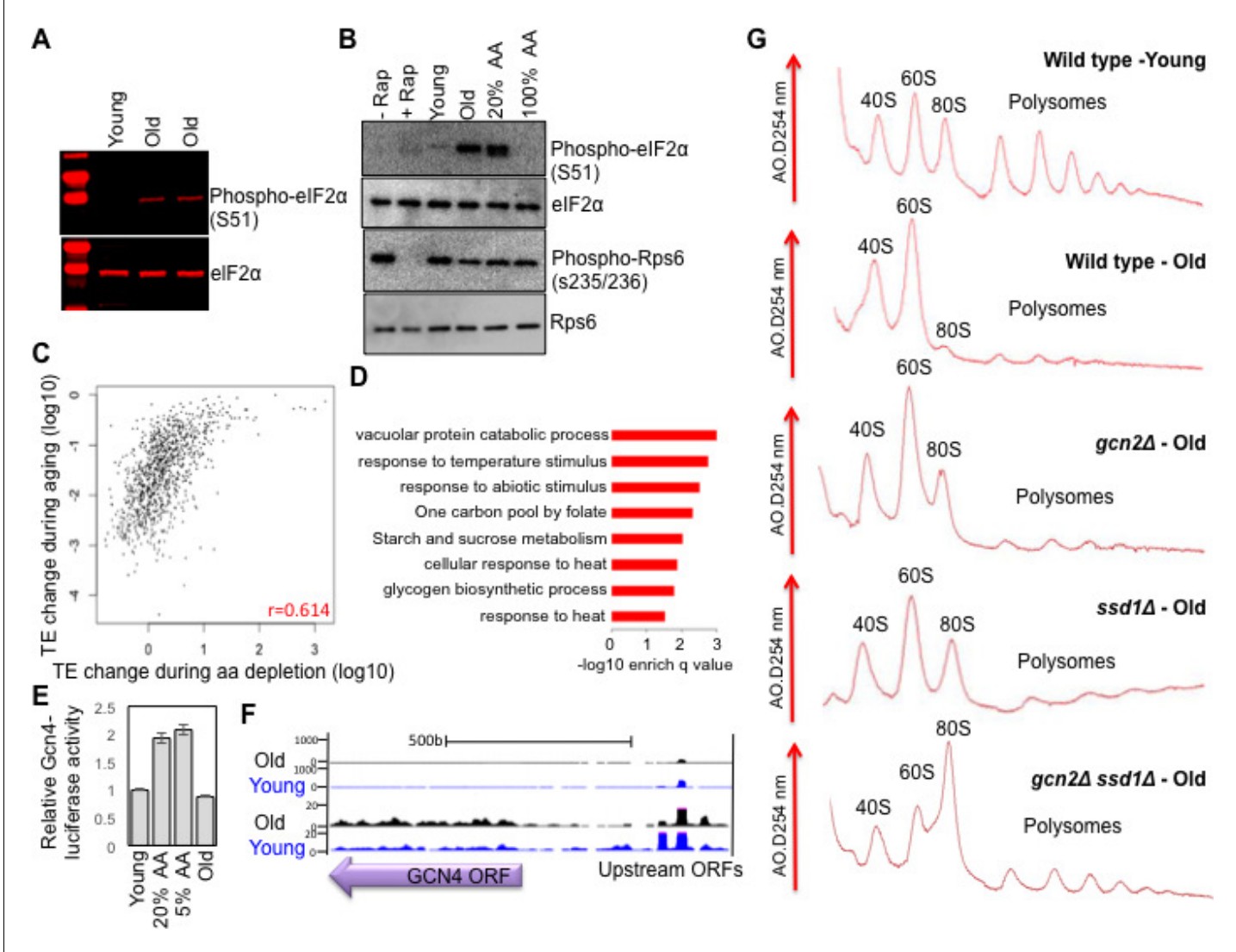

**Figure 4.** Activation of Gcn2 but not Gcn4 in old cells. (A) Western blot analysis detects the induction of eIF2α phosphorylation during aging. The equivalent number of cells worth of protein from young and old cells was loaded (strain ZYH2). Two independent samples are shown for old cells. (B) Induction of eIF2α phosphorylation but not inactivation of TOR during aging. The equivalent number of cells worth of protein from young and old cells was loaded, or cells treated with or without Rapamycin (200 nM for 2 hr) or with 100% or 20% of the normal level of amino acids (all in strain ZYH2). (C) Comparison of the changes in translational efficiency during aging vs. translational efficiency during amino acid depletion. The translation efficiency values have been sorted, grouped in sets of 5 and averaged; the x and y coordinates of each dot are the averages of the amino acid depletion and the aging translation efficiency values, respectively. (D) Genes that are highly translated only during starvation but not during aging are Gcn4-induced genes. David pathway analysis of genes with high increase in translational efficiency only in starvation but not in aging. (E) Gcn4 luciferase levels with either 20% or 5% of the normal level of amino acids (to induce Gcn4) in comparison to young and old yeast (strain ZYH2). Averages and standard deviation are plotted of three independent experiments. (F) UCSC browser screen shot of the distribution of ribosome footprint reads across the *GCN4* gene ORF and its uORFs. The bottom shows 50 fold lower numbers on the y-axis. Pink indicates values that are off scale. (G) UV traces of RNA distributions on sucrose gradients from old cells of young wild type cells (ZHY2 strain), old wild type cells, old *gcn2Δ* cells (ZHY8 strain), old *ssd1Δ* cells (ZHY7 strain), and old *ssd1Δ gcn2Δ* cells (ZHY22 strain) from polysome profiles ran in parallel. See also *Figure 4—figure supplements 1–3*.

DOI: https://doi.org/10.7554/eLife.35551.013

The following figure supplements are available for figure 4:

**Figure supplement 1.** Overlap analysis of the translational efficiency changes during aging vs amino acid depletion.

DOI: https://doi.org/10.7554/eLife.35551.014

**Figure supplement 2.** The genes with the lowest translational efficiency change during aging were not detectable in the analysis during amino acid depletion.

DOI: https://doi.org/10.7554/eLife.35551.015

*Figure 4 continued on next page*

*Figure 4 continued*

**Figure supplement 3.** Genes that are highly translated only in starvation but not in aging are GCN4 induced genes.

DOI: https://doi.org/10.7554/eLife.35551.016

of phosphorylation of eIF2α in old cells, the global reduction of translational initiation in old cells is similar to that seen during stress conditions that induce eIF2α phosphorylation (*Kimball, 1999*), with the major difference being that Gcn4 and its downstream targets are not upregulated in old cells.

## Ssd1 induction and Gcn2 activation both contribute to reduced protein synthesis in old cells

The experiments above indicated that Ssd1 is induced in old cells, and that overexpression of Ssd1 is capable of reducing protein synthesis (*Figure 1*). Meanwhile levels of phosphorylation of the Gcn2 target eIF2α, a known global repressor of translational initiation, are elevated in old cells (*Figure 4A,B*). To discern whether either of these mechanisms are responsible for the down-regulation of protein synthesis that occurs during replicative aging (*Figures 1* and *2*), we assessed whether deletion of *GCN2* or *SSD1* could restore protein synthesis in old cells. By specifically examining polysome profiles from old yeast, it was apparent that deletion of either gene *GCN2* or *SSD1* restored the monosome peak that is absent in the old wild type yeast (*Figure 4F*). Furthermore, deletion of both *GCN2* and *SSD1* further restored the protein synthesis in old cells (*Figure 4F*). These results indicate that Ssd1 induction and Gcn2 activity each contribute to the down-regulation of protein synthesis that occurs during replicative aging.

## Activation of Gcn2 by overexpression of a tRNA extends replicative lifespan, in a manner dependent on Gcn4

Extension of replicative lifespan by inactivating TOR or depleting ribosomes correlates with reduced protein synthesis, Therefore, we asked whether experimental activation of the Gcn2 kinase in young cells is sufficient to extend lifespan. Given that Gcn2 can be activated by uncharged tRNAs (i.e. tRNAs not bound to amino acids), we overexpressed *IMT4,* which encodes an initiator tRNA$^{MET}$. Overexperssion of *IMT4* from a 2μ vector creates uncharged tRNAs, and this led to a significant increase in replicative lifespan compared to isogenic cells carrying the empty 2μ vector using both a survival assay time course that measures the ability of the aging mothers to form colonies over a time course of the MEP (*Figure 5A*) and the micromanipulation assay (*Figure 5B*). tRNA overexpression resulted in Gcn2 activation in young cells, reflected by eIF2α phosphorylation (*Figure 5C*), but had no influence on Rps6 phosphorylation suggesting that it did not inactivate TOR (*Figure 5B*). This result shows that Gcn2 activation does not inhibit the TOR pathway in yeast, at least in rich media conditions. This is in contrast to the activation of Gcn2 that results from TOR inhibition (*Figure 5C*) (*Cherkasova and Hinnebusch, 2003*). Furthermore, tRNA overexpression induced the Gcn4-luciferase reporter to equivalent levels as amino acid starvation, in a manner that was dependent on Gcn2 (*Figure 5D*). By polysome analysis, we found that tRNA overexpression reduced overall protein synthesis in young cells (*Figure 5E*), consistent with induced eIF2α phosphorylation (*Figure 5C*).

To determine whether overexpression of the tRNA encoded by *IMT4* was indeed extending lifespan through the stress response pathway, we deleted the gene encoding the *GCN2* kinase. We found that the replicative lifespan extension that was achieved by *IMT4* overexpression was fully dependent on Gcn2 (*Figure 5F*). Notably, *GCN2* deletion alone had no effect on lifespan (*Figure 5F*), indicating that the reduced protein synthesis in old cells that is mediated by eIF2α phosphorylation (*Figure 4G*) does not optimize the normal lifespan.

Lifespan extension resulting from Gcn2 activation in young cells could be due to the global inhibition of translational initiation that results from eIF2α phosphorylation. Alternatively, the production of Gcn4, which occurs downstream of the global inhibition of translational initiation, and the many stress responses that Gcn4 transcriptionally regulates may be responsible for extending lifespan, or a combination of both. To differentiate between these possibilities, we deleted *GCN4* in the context of *IMT4* overexpression. We found that the replicative lifespan extension that was achieved by *IMT4* overexpression was dependent on *GCN4* (*Figure 5G*). These results show that tRNA overexpression

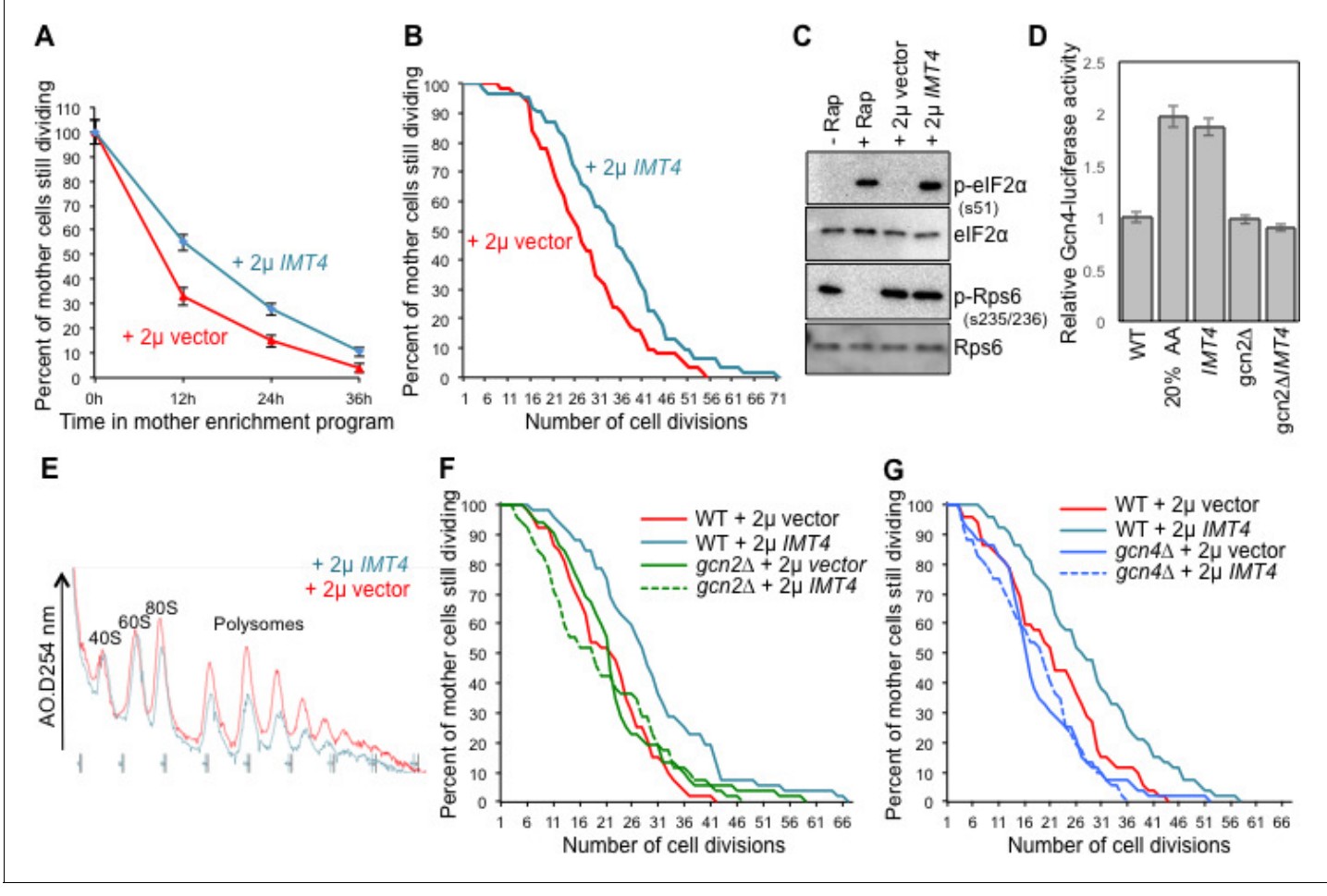

**Figure 5.** Overexpressing a tRNA activates Gcn2 and extends lifespan. (A) Replicative lifespan analysis by plating at the indicated times in the MEP, for the wild type cells (ZHY2) harboring the plasmid pRS423-*IMT4* or empty pRS423 vector. Average and standard deviation of three independent experiments are shown. (B) Replicative lifespan analysis for the wild type cells (ZHY2) harboring the plasmid pRS423-*IMT4* or empty pRS423 vector. (C) Comparison of eIF2α and Rps6 phosphorylation in the same strains as used in B. Wild type cells with 200 nm rapamycin treatment for two hours or without treatment are shown as controls. (D) Expression of Gcn4-luciferase fusion with 20% of the normal level of amino acids (to induce Gcn4) in comparison to wild type (ZHY2) or *gcn2Δ* cells (ZHY8) harboring the plasmid pRS423-*IMT4* or empty pRS423 vector as indicated. Averages and standard deviation are plotted of three independent experiments. (E) Polysome profiling using the same number of cells from the strains used in A. (F) Replicative lifespan analysis for BY4742 or BY4742*gcn2Δ* strains harboring the plasmid pRS423-*IMT4* or empty pRS423 vector. 2-tail Mann-Whitney test: WT +2μ vector vs WT +2μ *IMT4* p=0.00072; all other differences are not significant. (G) Replicative lifespan analysis for BY4742 or BY4742*gcn4Δ* harboring the plasmid pRS423-*IMT4* or empty pRS423 vector. 2-tail Mann-Whitney test: WT +2μ vector vs WT +2μ *IMT4* p=0.000614; all other differences are not significant.

DOI: https://doi.org/10.7554/eLife.35551.017

activates Gcn2 to extend replicative lifespan, in a manner seemingly independent of TOR inactivation, but dependent on Gcn4.

## Lifespan extension by Gcn4 overexpression is dependent on autophagy

Given that Gcn4 was required for lifespan extension that results from *IMT4* overexpression (*Figure 5G*), we asked if overexpression of Gcn4 was sufficient to extend yeast replicative lifespan, using a strain in which the inhibitory uORFs had been deleted, while keeping *GCN4* transcription under the control of the endogenous promoter (*Hinnebusch, 1985*). In comparison to an isogenic control, overexpression of Gcn4 significantly extended lifespan (*Figure 6A*). Importantly, overexpression of Gcn4 led to no apparent induction of Gcn2 activity or inhibition of TOR activity (*Figure 6B*). As such, Gcn4 translation is sufficient to extend replicative lifespan in a manner seemingly

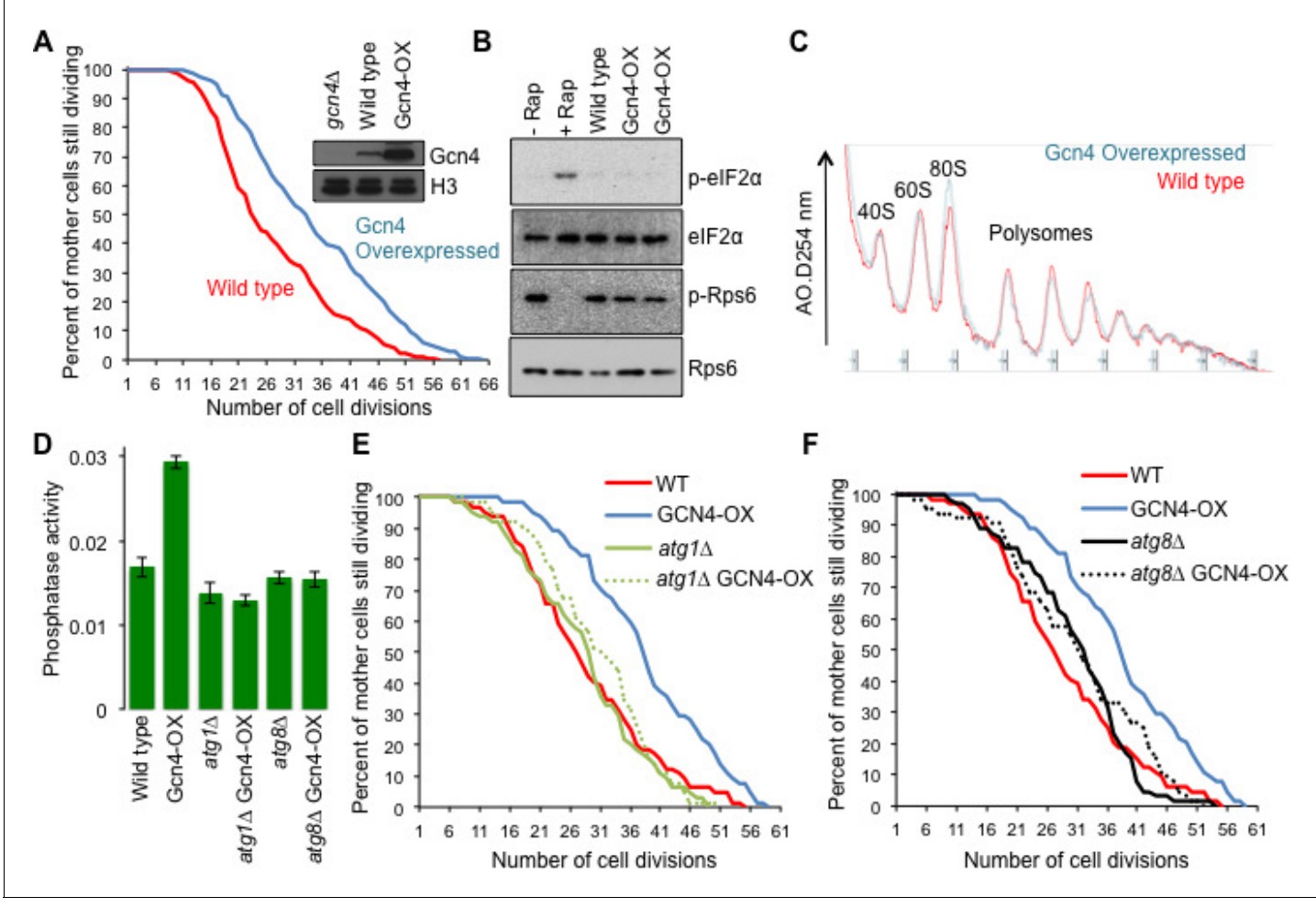

**Figure 6.** Overexpressing Gcn4 extends lifespan in a manner dependent on autophagy, without reducing global protein synthesis. (A) Replicative lifespan analysis for the wild type cells (strain F113) or with integration of the Gcn4 uORF deletion mutant (ZHY12). 2 tailed Mann-Whitney test p<0.00001. Inset shows Gcn4 protein levels in the indicated strains. (B) As for *Figure 5C*, using the strains used in A. (C) As for *Figure 5E*, using the strains used in A. (D). Analysis of autophagy using strains: WT (WLY176), GCN4-OX (ZSY2119), *atg1Δ* (WLY192), *atg1Δ* GCN4-OX (ZSY21113), *atg8Δ* (ZSY2117) and *atg8Δ* GCN4-OX (ZSY2118). (E) Replicative lifespan analysis for the indicated isogenic strains listed in D. 2 tailed Mann-Whitney tests: WT vs GCN4-OX p<0.00001. The difference between WT, *atg1Δ* and *atg1Δ* GCN4-OX were not significantly different. (F) Replicative lifespan analysis for the indicated isogenic strains, listed in D. 2 tailed Mann-Whitney tests: WT vs GCN4-OX p<0.00001. The difference between WT, *atg8Δ* and *atg8Δ* GCN4-OX were not significantly different.

DOI: https://doi.org/10.7554/eLife.35551.018

independent of TOR inhibition. Gcn4 has been reported to repress genes required for protein synthesis (*Mittal et al., 2017*), but we observed no suppression of global protein synthesis in polysome profiling analysis of young cells overexpressing Gcn4 (*Figure 6C*). These results indicate that Gcn4 overexpression is sufficient to extend replicative lifespan in the absence of reduced global protein synthesis.

Gcn4 activates expression of a variety of genes that mediate amino acid biosynthesis, purine biosynthesis, organelle biosynthesis, ER stress response, mitochondrial carrier proteins and autophagy (*Pakos-Zebrucka et al., 2016*), any of which could be responsible for the lifespan extending influence of Gcn4 overexpression. Given the many links between autophagy and aging in other species (*Rubinsztein et al., 2011*), we asked whether induction of Gcn4 activates autophagy in rich growth conditions. To do this, we used the quantitative Pho8Δ60 assay of nonspecific autophagy (*Figure 6D*). To determine whether the lifespan extension that resulted from Gcn4 overexpression required autophagy we deleted *ATG1*. While *ATG1* deletion alone did not have a significant effect on replicative lifespan compared to wild type, deletion of *ATG1* completely blocked lifespan

extension by Gcn4 overexpression (*Figure 6E*). To provide additional evidence for the requirement of autophagy in the lifespan extension mediated by Gcn4 activation, we deleted *ATG8* (*Figure 6F*). The lifespan of the wild type, *atg8Δ* and *atg8Δ GCN4-OX* strains were not statistically different, revealing that the lifespan extension caused by Gcn4 overexpression requires *ATG8*. Taken together, these data demonstrate that Gcn4 induction extends replicative lifespan in a manner that is fully dependent on autophagy (*Figure 7*).

## Discussion

Here we present a comprehensive analysis of protein synthesis on a genome-wide scale during yeast replicative aging. This reveals that there is a global defect in the initiation of protein synthesis with increasing replicative age. We have also uncovered two molecular mechanisms that contribute to the global decline in protein synthesis during normal aging: activation of the Gcn2 kinase and induction of the Ssd1 mRNA binding protein in old cells. Physiological down-regulation of protein synthesis by Ssd1, but not Gcn2, during aging is a mechanism that maximizes lifespan. In young cells, the experimental induction of Ssd1 extends lifespan. Meanwhile, activation of Gcn2 in young cells extends lifespan in a manner dependent on Gcn4, where Gcn4 induction is sufficient to extend replicative lifespan in an autophagy-dependent manner. Taken together, these data show that lifespan can be extended by experimentally enhancing mRNA transport to aggregated P-bodies, or via enhancing activation of the stress response, which functions via induction of Gcn4 and subsequent autophagy (*Figure 7*).

### Reduced protein synthesis during yeast replicative aging

Polysome profiling and ribosome profiling revealed a clear defect in translational initiation in middle aged cells, that was even more so in old cells (*Figures 1* and *2*). The genes that had the most profound reduction in translation efficiency during aging were strikingly enriched in encoding proteins involved in ribosome biogenesis or the regulation of protein synthesis. This may be because they are among the most highly translated genes normally given that their encoding mRNAs are so abundant

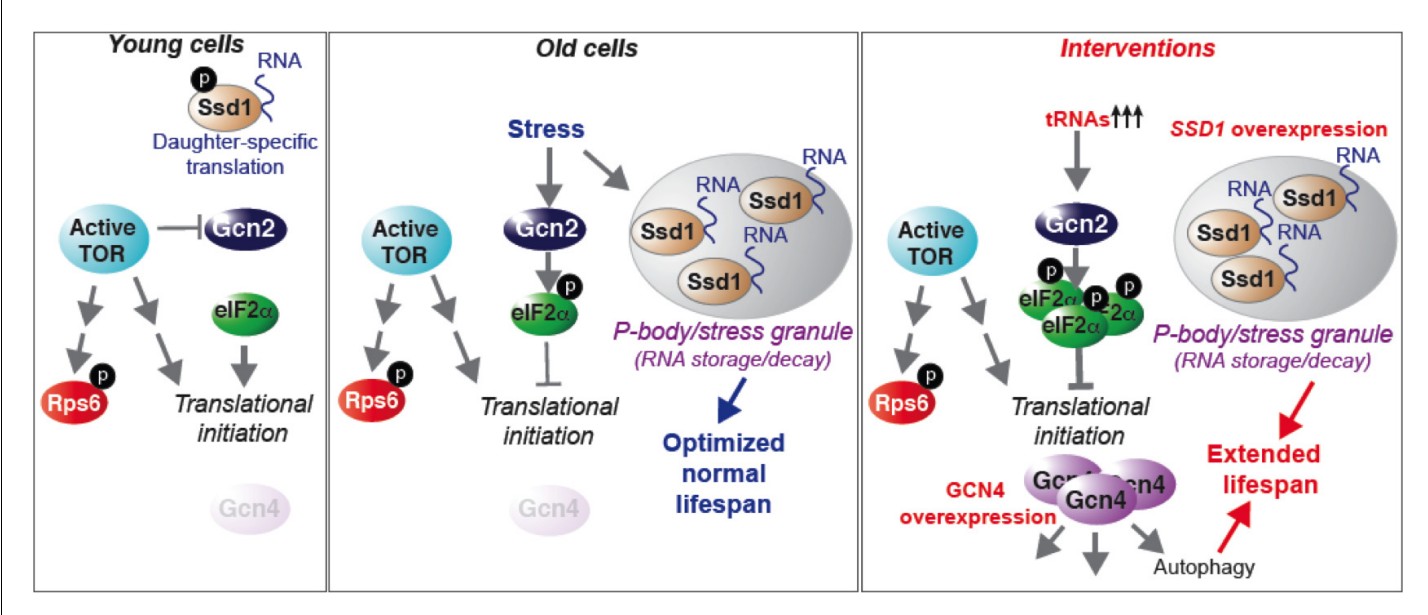

**Figure 7.** Model summarizing our results, as described in the text. See also *Figure 7—figure supplement 1*.
DOI: https://doi.org/10.7554/eLife.35551.019
The following figure supplement is available for figure 7:

**Figure supplement 1.** Amino acid depletion induces P bodies and induces Ssd1.
DOI: https://doi.org/10.7554/eLife.35551.020

(*Figure 3*), so are more sensitized to reduction in translational initiation. Alternatively there may be a positive feedback loop to more strongly repress translation of the transcripts encoding the translation machinery under conditions of reduced protein synthesis. Strikingly, no genes had significantly increased translational initiation in old cells, and those that had no change were expressed at very low levels in young cells. Inhibition of the TOR pathway does not appear to be involved in the reduction in protein synthesis during aging, given that phosphorylation of Rps6 persists throughout aging (*Figure 4B*). To explain the reduced translational initiation in old cells that was apparent from the Ribo-seq analysis, as opposed to reduced elongation or increased pausing, we have deciphered two mechanisms at play, as discussed below.

## P-body aggregation and Ssd1 function in translational repression during aging and lifespan extension

Our work implicates Ssd1's role in repressing translation as a regulator of longevity. The initial association between Ssd1 and mRNAs occurs within the nucleus, apparently during transcription (*Kurischko and Broach, 2017*), potentially via an interaction between its aggregation-prone prion-like domain and the phosphorylated C-terminal domain (pCTD) of RNA polymerase II (*Phatnani and Greenleaf, 2004*). Ssd1 is best known for delivering specific mRNAs to the daughter cell for their asymmetric translation. However Ssd1 can also take mRNAs to P-bodies, the sites of mRNA decay and storage of translationally-repressed mRNAs (*Kurischko et al., 2011*). In response to cellular stresses that generally block translational initiation, such as glucose depletion, hypertonic stress and heat shock, P-bodies aggregate and stress granules form (*Kedersha and Anderson, 2009*). Furthermore, stress (induced by glucose starvation, a known longevity extending intervention) causes Ssd1, and presumably its mRNA cargo, to localize to P-bodies and stress granules (*Kurischko et al., 2011*). In agreement, during aging we find that there is a huge increase in the proportion of old cells with visible P-bodies and find that Ssd1 localizes to these age-induced P-bodies. Our observations are unlikely to be specific to yeast, given that P-bodies accumulate in an age-dependent manner in the somatic cells of adult worms (*Rousakis et al., 2014*).

The role of Ssd1 in delivering mRNAs to the daughter cell for translation is promoted by phosphorylation of Ssd1 by the Cbk1 kinase (*Jansen et al., 2009*; *Kurischko et al., 2011*). Meanwhile, the localization of Ssd1 to aggregated P-bodies for translational repression is normally inhibited in non-stressed cells, at least partly by the phosphorylation of Ssd1 by Cbk1 (*Jansen et al., 2009*; *Kurischko et al., 2011*). Indeed polysome profiles of *SSD1* vs. *ssd1Δ* young cells are indistinguishable, showing no increased translation upon loss of Ssd1 (*Jansen et al., 2009*). However, we found that moderate overexpression of Ssd1 distinctly reduced protein synthesis in young cells (*Figure 1E*), and extended replicative lifespan (*Figure 1G*). Meanwhile, loss of Ssd1 shortened replicative lifespan (*Figure 1F*) and increased translation in old cells (*Figure 4G*). It was somewhat surprising that moderate overexpression of Ssd1 globally reduced protein synthesis in young cells (*Figure 1E*), given that Ssd1 has only been reported to bind to 59 and 152 specific mRNAs (*Jansen et al., 2009*; *Hogan et al., 2008*). Furthermore, these were not particularly abundant mRNAs. However, identification of the Ssd1-bound mRNAs was performed in non-stressed conditions, where Ssd1 normally binds to specific mRNAs that are targeted for daughter-specific translation. It is possible that the loss of Cbk1 phosphorylation and/or the elevated levels of Ssd1 that result from moderate Ssd1 overexpression, as seen in old cells (*Figure 1B*) or achieved experimentally (*Figure 1E*), may enable Ssd1 to interact with a wider variety of mRNAs, taking them to P-bodies to mediate global reduction in protein synthesis. In agreement, moderate overexpression of Ssd1 causes localization of Ssd1 to aggregated P-bodies under non-stress conditions, indicating that this overwhelms regulatory Cbk1 phosphorylations (*Kurischko et al., 2011*). Our data indicate that the transcriptional up-regulation of Ssd1 that occurs in old cells, and the accompanying translational repression caused by sequestering mRNAs to P-bodies, is a mechanism to decrease protein synthesis in order to maximize longevity (*Figure 7*).

In addition to promoting normal longevity, Ssd1-mediated retention of mRNAs in P-bodies as a mechanism to down-regulate protein synthesis may be relevant to known interventions that extend lifespan. Glucose depletion, one of the best known mechanisms to promote longevity, causes formation of P-bodies containing Ssd1 (*Kurischko et al., 2011*). Amino acid depletion, another intervention known to extend yeast replicative lifespan (*Lee et al., 2014*; *Jiang et al., 2000*) also led to P-body formation in young yeast (*Figure 7—figure supplement 1*), as it does in mammalian cells

(*Aizer et al., 2014*). As such, it will be interesting to determine if extension of lifespan by calorie restriction and amino acid depletion depends on Ssd1-mediated repression of translation.

Ssd1 could also account for asymmetric differences in protein synthesis, where mother cells have reduced protein synthesis, while daughter cells maintain efficient protein synthesis. The Cbk1-mediated phosphorylation of Ssd1 regulates the cellular destination of Ssd1 and its bound mRNAs. Inhibition of Cbk1 or stress (glucose) causes Ssd1 to localize to P-bodies. Similarly, moderate overexpression of Ssd1 causes Ssd1 to localize to the stress granules in mothers but not buds that will become daughters (*Kurischko et al., 2011*). Considering that most Cbk1 localizes asymmetrically to buds (*Colman-Lerner et al., 2001*), our data are consistent with the model that moderate Ssd1 overexpression overwhelms regulatory Cbk1 phosphorylations in mother cells where Cbk1 concentrations are low, but not in buds where Cbk1 concentrations are higher. The mechanism that specifically upregulates Ssd1 transcription in old cells, beyond histone depletion, remains to be determined. Moreover, Ssd1 translation efficiency decreased during aging (Table S1), suggesting that mechanisms may exist to stabilize Ssd1 protein in old cells. Noteworthy, amino acid depletion, a lifespan extending intervention, is also accompanied by increase in levels of Ssd1 protein (*Figure 7— figure supplement 1*). While the direct sequence counterpart of Ssd1 in multicellular eukaryotes is not obvious, Ssd1 functions analogously to hnRNPs, which bind mRNA co-transcriptionally, are exported to the cytoplasm and target mRNAs to sites of localized translation and P-bodies (*Geuens et al., 2016*). Future studies will no doubt uncover the functional equivalent of Ssd1 in mammalian cells, and its moderate induction or dephosphorylation might be a favorable target for interventions to extend lifespan in mammals, via its ability to repress translation.

## Gcn2 activity during aging down-regulates translation in old cells without affecting lifespan

Significant phosphorylation of eIF2α occurred in old cells, indicating that Gcn2 is likely activated during aging (*Figure 4A,B*). While reduced dephosphorylation of eIF2α could also contribute to our results in old cells, this would still depend on Gcn2-mediated eIF2α phosphorylation, which doesn't usually occur in cells grown in rich media. The activation of Gcn2 during aging is not merely due to nutrient depletion during aging because we refresh media during the MEP time course. The Gcn2-mediated phosphorylation of eIF2α is in agreement with the global decrease in translational initiation in old cells revealed by Ribo-seq analysis (*Figure 2*). Despite Gcn2 being activated in old cells, this did not result in Gcn4 activation (*Figure 4D–F*). This is in agreement with deletion of *GCN4* having negligible effects on replicative lifespan (*Managbanag et al., 2008*) (*Figure 5G*). There may be a threshold of Gcn2 activation that needs to be crossed to achieve sufficient reduction in global protein synthesis to trigger ribosome scanning from the upstream ORFs to the *GCN4* ORF, alternatively, the normal mechanisms of Gcn4 translation that occur by ribosome scanning may be non-functional in old cells. Regardless, the Gcn2 activation in old cells clearly contributes to the decrease in protein synthesis in old cells because *GCN2* deletion partially restores protein synthesis in old cells (*Figure 4G*). However, deletion of *GCN2* does not affect lifespan (*Figure 5F*), indicating that the reduction in protein synthesis that occurs in old cells due to eIF2α phosphorylation does not maximize normal lifespan. This is in contrast to the Ssd1-mediated down-regulation of protein synthesis during aging. It will be interesting to determine in the future whether Ssd1 and Gcn2 repress synthesis of different proteins during aging that may differentially affect longevity.

The question remains, what kind of stress is activating Gcn2 in old cells? Although Gcn2 is activated by uncharged tRNAs, Gcn4 synthesis, resulting presumably from Gcn2 activation, is induced under many conditions besides amino acid depletion. These include starvation for purines (*Rolfes and Hinnebusch, 1993*), glucose limitation (*Yang et al., 2000*), ER stress (*Deloche et al., 2004*), growth on non-fermentable carbon sources like ethanol (*Yang et al., 2000*), high salt (*Goossens et al., 2001*), treatment with alkylating agents (*Natarajan et al., 2001*) and TOR inhibition (*Cherkasova and Hinnebusch, 2003*; *Kubota et al., 2003*). All these conditions that activate Gcn4 depend on the Gcn2 kinase and many extend yeast replicative lifespan (*Postnikoff et al., 2017*). Future studies will reveal which of these stresses activate Gcn2 in old cells.

By contrast to the Gcn2 activity that is detected during aging that does not induce Gcn4 and is not required for normal lifespan (*Figure 4*), experimental activation of Gcn2 by tRNA overexpression induced Gcn4 and extended lifespan, in a manner dependent on Gcn4 (*Figure 5*). The correlation between induction of Gcn4 and lifespan extension seen in our work is reminiscent of the Gcn4

dependence for full yeast lifespan extension that resulted from depletion of large ribosomal subunits, *TOR1* deletion, or dietary restriction (*Steffen et al., 2008*).

## Intertwined functions of Ssd1 and Gcn2 during aging

It is possible that the reduced protein synthesis that results from induction of Ssd1 and Gcn2 activation within old cells, although occurring via distinct mechanisms, may be linked. P-body aggregation is known to be mediated by accumulation of stalled preinitiation complexes, especially 40S ribosomes and mRNA (*Panas et al., 2016*). Inhibition of translation by Gcn2-mediated eIF2α phosphorylation during aging would lead to accumulation of 40S ribosomes and mRNA, which could trigger aggregation of P-bodies in old cells. As such, it is feasible that the two mechanisms that we observe here are interrelated. That is, the inhibition of translation by Gcn2 activation may promote the aggregation of P-bodies in old cells. P-body proteins in turn bind to Ssd1 within the nucleus during transcription, collecting Ssd1 and delivering it to P-bodies, along with its mRNA cargo (*Kurischko and Broach, 2017*). Moreover, P-body proteins affect general translation repression. Yeast strains lacking P-body components are defective in the global translation repression that occurs in response to glucose deprivation or amino acid starvation (*Coller and Parker, 2005*). Conversely, overexpression of the P-body components Dhh1p or Pat1p leads to global inhibition of translation in yeast and an increase in P-bodies (*Coller and Parker, 2005*). Thus, the mRNAs found in P-bodies should be considered part of a mechanism of translational repression acting on the majority of cytoplasmic mRNAs and targeting them for translation repression and/or degradation. Our data indicates that the reduction in protein synthesis during aging is mediated partly by increased eIF2α phosphorylation dependent on Gcn2 and by Ssd1-dependent translational repression, and that these mechanisms have at least some non-overlapping functions in reducing protein synthesis during aging, given that the *gcn2 ssd1* double mutant had the least translation repression in old cells (*Figure 4G*). Clearly, these two mechanisms that reduce protein synthesis in old cells have different influences on the normal aging process, as seen by the lifespan shortening upon Ssd1 loss but not Gcn2 loss.

Additional factors also clearly contribute to the reduction in protein synthesis that occurs during aging. For example, cells get bigger as they age (*Janssens and Veenhoff, 2016*), diluting the effective cellular concentration of ribosomes, which may further reduce protein synthesis. This might be exacerbated by the simultaneous increase in total mRNA levels during aging (*Hu et al., 2014*). Indeed, an elegant analysis of ribosome concentrations in individual old cells found that those cells with the lowest ribosome concentration will go on to have the longest lifespan (*Janssens and Veenhoff, 2016*). Without rigorous and direct measurements of elongation transit times during aging, it is feasible that alterations in elongation may also contribute to the reduction in protein synthesis that occurs during aging.

## Why would reduced protein synthesis maximize normal longevity?

A growing body of studies indicates that experimental attenuation of protein biosynthesis significantly extends lifespan in many model organisms (*Tavernarakis, 2008*). Furthermore, the natural decline in protein synthesis during replicative aging that we have uncovered also appears to be beneficial for optimization of normal lifespan, suggested by the shortened lifespan upon inactivating Ssd1. The potential reasons why reduced protein synthesis could be beneficial for maximizing longevity are multifold. Translation is one of the most energy-consuming cellular processes and it requires an estimated 50% of the total cellular energy. Consequently reduced protein synthesis would result in notable energy savings. This energy could then be diverted to cellular repair and maintenance processes, thus promoting organismal longevity. Noteworthy, there is no evidence yet for energy being a limiting factor during aging, although a greater need for the mitochondria to generate more energy via oxidative phosphorylation would lead to more oxidative stress, which could be deleterious for lifespan. Protein homeostasis would also be better preserved when mRNA translation is reduced. For example, age-associated loss of protein homeostasis due to misfolded, damaged and aggregated proteins, all of which occur during aging, could be reduced by producing less proteins which would prevent overloading the unfolded protein response. It is presently unclear which of these pathways, or others, holds the true answer to why reduced protein synthesis enables lifespan extension. On the flip side, protein synthesis is an essential process and its complete

inhibition is lethal, indicating that a delicate balance exists between obtaining the maximal longevity benefit and the detrimental impairment of protein metabolism. Inhibition of the TOR pathway by mutations or drugs has a well-appreciated ability to extend lifespan, but whether this requires the accompanying reduction in protein synthesis is still unknown.

## Induction of Gcn4 extends lifespan, in an autophagy-dependent manner

Gcn4 is required for several different interventions that extend replicative lifespan. Deletion of *GCN4* reduces the yeast replicative lifespan extension that results from deletion of *LOS1*, a tRNA exporter, and from deletion of *AFG3*, which is involved in mitochondrial mRNA translation (*McCormick et al., 2015*; *Delaney et al., 2013*). Replicative lifespan can also be extended by depletion of the 60S ribosomal subunits in yeast (*Steffen et al., 2008*), which also induces Gcn4 translation (*Mittal et al., 2017*) and deletion of *GCN4* reduces this lifespan extension (*Steffen et al., 2008*). Dietary restriction and TOR inhibition also increase Gcn4 translation, while *GCN4* deletion reduces the ability of TOR inactivation and dietary restriction to extend replicative lifespan (*Steffen et al., 2008*). In agreement, we find that the lifespan extension due to tRNA overexpression is dependent on Gcn4 (*Figure 5*). These results suggest that it is the translation of Gcn4 indirectly resulting from repression of global translational initiation that is key for lifespan extension. In addition to inducing the genes that mediate amino acid biosynthesis, purine biosynthesis, organelle biosynthesis, ER stress response, mitochondrial carrier proteins and autophagy (*Pakos-Zebrucka et al., 2016*), Gcn4 was recently shown to repress the transcription of the translation machinery (*Mittal et al., 2017*). However, it was unknown whether this Gcn4-mediated repression of the translation machinery contributes to the role of Gcn4 in lifespan extension.

The question remained, which of the stress responses activated by Gcn4 lead to replicative lifespan extension? Overexpression of Gcn4 from an *ADH1* promoter was very recently shown to extend replicative lifespan and reduce bulk protein synthesis (*Mittal et al., 2017*). By contrast, we found that overexpression of Gcn4 from its own promoter robustly extends replicative lifespan, but does not reduce protein synthesis (*Figure 6*). Both approaches deleted the inhibitory uORFs that normally promote disassembly of ribosomes before they reach the *GCN4* ORF; hence uORF deletion promotes Gcn4 translation. The distinct effects on protein synthesis could be due to distinct expression levels of Gcn4 from the different promoters, where *ADH1* would presumably lead to strong constitutive Gcn4 expression. Alternatively, the difference could be due to the fact that our study measured endogenous protein synthesis by polysome profiling, while Mittal et al., measured incorporation of an exogenous labeled glycine analog into bulk proteins (*Mittal et al., 2017*) which might be diluted out upon the synthesis of endogenous glycine that is induced by Gcn4 overexpression. Regardless, the fact that we observed replicative lifespan extension by overexpressed Gcn4 in conditions that did not reduce global protein synthesis suggests that other pathways regulated by Gcn4 may be more central for extending lifespan. Indeed, we found that inactivation of autophagy fully negated the lifespan extending effect of Gcn4 overexpression (*Figure 6E,F*).

Autophagy has been extensively linked to lifespan extension regimens previously. Although this has mostly been from studies in metazoans or during chronological aging (*Rubinsztein et al., 2011*; *Tyler and Johnson, 2018*), a recent study during yeast replicative aging demonstrated that autophagy was required for lifespan extension in response to mutations that induce ER stress (*Ghavidel et al., 2015*). The TOR pathway and the RAS/PKA pathway inhibit autophagy when nutrients are abundant. However, when nutrients are limiting, the nutrient sensing pathways are inactivated and their inhibitory influence on autophagy is released. Our data, however, suggests shows that overexpression of Gcn4 is sufficient to activate autophagy without the apparent need for inactivation of the TOR pathway (*Figure 6D,B*).

In summary, we find that protein synthesis declines naturally during the normal replicative aging process, in a manner that is not apparently accompanied by inhibition of the TOR pathway. Instead, the reduction in protein synthesis during aging is mediated via activation of the Gcn2 kinase, which inhibits translational initiation through eIF2α, in combination with Ssd1-mediated delivery of mRNAs to aggregated P-bodies. The fact that experimental induction of Ssd1 or activation of Gcn2, or overexpression of Gcn4, extends replicative lifespan in an autophagy-dependent manner, demonstrates the utility of these pathways as targets for longevity enhancing interventions.

## Materials and methods

### Yeast culture, Strains and Plasmids

Cells were grown under standard conditions, unless otherwise noted. Yeast strains, plasmids, and oligonucleotides are listed in *Supplementary file 1*. Epitope-tagged sORFs were generated using homologous recombination (*Longtine et al., 1998*) or standard molecular cloning strategies. For isolating old cells, we used the MEP (*Lindstrom and Gottschling, 2009*) with the previously described slight modifications (*Hu et al., 2014*). Middle-aged cells were isolated after 15 hr in estradiol while old cells were isolated after 30 hr in estradiol. Biotin-affinity was used to separate the old or middle-aged cells from the arrested young cells. Yeast were grown in YEP +2% glucose or 0.5% galactose. For amino acid depletion, yeast cells were grown to mid log phase in synthetic medium with 100% of the normal amino acids mixture, and then cells were shifted to the SC medium with 20% of the normal amino acids mixture for 5 hr and cells were collected for further analysis. For the plasmid pZH1 construction, pRS416 *GFP-ATG8* plasmid was digested with EcoRI and SpeI, and the fragment containing *GFP-ATG8* was ligated with EcoRI and SpeI into the same sites in the pRS306 vector. Plasmids are listed in *Supplementary file 1*.

### Western blotting

Strains were grown in the appropriate medium. Cells were pelleted and re-suspended in yeast 2X SDS sample buffer (2 ml Tris (1M, pH6.8), 4.6 ml glycerol (50%), 1.6 ml SDS (10%), 0.4 ml bromophenol blue (0.5%), 0.4 ml β-mercaptoethanol) and boiled. Crude extract were resolved by SDS page and a rabbit polyclonal antibody to the HA tag (ab9110, Abcam) was used to detect the HA-tagged proteins; a rabbit monoclonal antibody specific for anti-eIF2α was generously provided by Dr. Thomas Dever (NIH, Bethesda, MD). A rabbit polyclonal anti-EIF2S1 (Phospho S51) (#2211 s, Cell Signalling) was used to detect the phosphorylated eIF2α. A rabbit polyclonal antibody anti-phospho-S6 ribosomal protein (S235/236) was used to detect the endogenous levels of ribosomal protein S6 only when phosphorylated at serine 235 and 236. A rabbit polyclonal antibody anti-Rps6 (ab40820, Abcam) was used to detect the endogenous level of ribosome protein S6. A rabbit polyclonal antibody Anti-Rad52 (Sc-50445, Santa Cruz Biotechnology) was used as a loading control. A mouse monoclonal antibody against Gcn4 (Ab00436-1.1, Absolute Antibody) was used to detect Gcn4. A rabbit polyclonal antibody against Histone H3 (ab1791, Abcam) was used as a loading control. IRDye Anti-rabbit secondary antibody was used that fluoresced at 700 nm. Membranes were scanned with a LiCor Odyssey scanning system and their software was used to quantitate the bands, using the average density method. Results shown are representative of results that were obtained multiple independent times. Gcn4-luciferase activity was measured as described previously (*Feser et al., 2010*).

### Fluorescence microscopy

Cells were grown to an $OD_{600}$ of 0.5–0.8 in YPD. For observation, cells were washed once and resuspended in synthetic medium (SC) supplemented with amino acids and glucose and immediately observed. For in vivo 4,6-diamidino-2-phenylindole (DAPI, Sigma) staining, cells were harvested by centrifugation and resuspended in Vectasheld Mount Medium for Fluorescence with DAPI (H-1200). All images are a *z*-series compilation of 6–10 images in a stack. For each condition, more than 100 cells were examined in 3 biologically independent experiments.

### Autophagy assay

Autophagic flux was assessed using the alkaline phosphatase-based assay that measures the delivery and subsequent activation in the vacuole of an altered form of the Pho8Δ60 phosphatase (*Noda et al., 1995*; *Scott et al., 1996*; *Noda and Klionsky, 2008*). Cells were grown in YPD medium and collected in log phase for the assay. The data presented here are the average of at least three independent experiments.

### Polysome profiling

Yeast strains were grown in YEPD, harvested at $OD_{600}$ of 0.8 after a 5 min cycloheximide treatment, and lysed by vortexing with glass beads. The crude ribosomal extracts from yeast cells were

separated by 10–50% sucrose gradient. The gradients were fractionated using a piston gradient fractionator (BioComp Instruments, Fredericton, NB, Canada) and UV absorbance at 254 nm was monitored using a UV-Monitor (BioRad, Hercules, CA).

### Ribosome footprinting

For ribosome footprinting, the clarified lysate from the same number of young and old cells were digested with RNase I and separated through a 10.5 ml 10–50% sucrose gradient. 80S monosome fractions were collected and the RNA was extracted following the same procedure as for global footprints. The TruSeq Ribo Profile kit RPYSC12116 (formerly ARTseq) was used to sequence ribosome-protected mRNA fragments. The Ambion ERCC RNA spike-in control (#4456740, Life Technology) was sonicated to reach an average size of 28nt and included in the library construction.

### RNA-Seq and library preparation

Total RNA isolation was performed using the MasterPure Yeast RNA Isolation Kit from Epicentre Biotechnologies (# MPY03100) following the manufacturer's instructions. All RNA fragments were converted into deep-sequencing libraries using TruSeq Stranded Total RNA Library Prep Kit with Ribo-Zero Gold (#RSS-122–2301, Illumina). The Ambion ERCC RNA spike-in control (#4456740, Life Technology) was included in the library construction.

### Mapping and analysis of Deep-sequencing data

Barcoded libraries were pooled and sequenced on an Illumina Hi-Seq2000 (ribosome footprints and RNA-seq). Reads were parsed into appropriate libraries by 50 barcode, and then adaptor sequences were removed. Trimmed reads were filtered and the remaining reads were mapped to the yeast sacCer3 genome, and viewed on the UCSC genome browser. Uniquely mapping reads 28 nt of ribosome footprints or RNA-seq reads were used for all analyses unless otherwise indicated.

Data analyses were performed according to the schematic diagram shown in *Figure 2—figure supplement 3*. Raw reads from all biological RNA-Seq and Ribo-Seq replicates were mapped to the *Saccharomyces cerevisiae* genomic annotation, version sacCer3 and assigned to genes using TopHat (*Trapnell et al., 2010*). The list of known genes was downloaded from the UCSC Table Browser (http://genome.ucsc.edu/cgi-bin/hgTables) and gene annotations were also obtained from UCSC. The sacCer3 and ERCC spike-in reference files were combined and processed into a TopHat index file using bowtie (*Langmead et al., 2009*). Reads counts for each gene were calculated by the htseq-count function in the tool HTSeq and subsequently, Fragments Per Kilobase of transcript per Million mapped reads (FPKM) were calculated and normalized using mean and quantile normalization procedures in R (https://www.r-project.org/). These values were then subjected to MEV (http://mev.tm4.org) to plot the heatmap figures. Translation efficiency change values were calculated in an identical manner to (*Ingolia et al., 2009*) and the comparison plot correlation presented in *Figure 4C* were realized in R. A similar comparison between our results and (*Ingolia et al., 2009*), was created in R and it shows significant overlap between the two rankings: the p value was computed using the Fisher exact test in R. The functional enrichment in gene sets was analysed based on DAVID Bioinformatics Resources (*Huang et al., 2007*), for which we used the Functional Annotation Chart function to retrieve a combined list of enriched functional terms. The negative log of the enrichment Q value (FDR-adjusted P value) was the sorting criteria for the top 25 terms for the top 500 genes by translation efficiency loss and the top 25 terms for the bottom 500 genes by translation efficiency loss. Internal transcription initiation information was obtained using the DANPOS (*Chen et al., 2013*) profile function with gene body normalization displaying the genomic coordinates against the values of the averaged signal. The rates of change of the young versus old samples were compared by fitting regression lines and by performing z-tests for the regression slopes in R. Visualizations of genomic data tracks were obtained using the USCS GenomeBrowser. (https://genome.ucsc.edu/). BEDTools (*Feser et al., 2010*) and FASTX-Toolkit (http://hannonlab.cshl.edu/fastx_toolkit) were also used for data processing.

### Replicative lifespan measurement

Replicative lifespan of virgin mother cells was determined by micromanipulation as previously described with minor modifications (*Kennedy et al., 1994*). Cells were streaked onto YEPD plates,

and a single colony was selected for life span analysis. Cells were incubated nightly at 12℃ to impede division. Yeast were initially grown on YEP +2% glycerol plates to eliminate yeast cells lacking mitochondria. At least 50 mother cells were examined for each strain. Replicative lifespan analysis during the MEP was determined using the colony growth assay after adding estradiol as described earlier (*Lindstrom and Gottschling, 2009*).

## Data availability

Next generation sequencing data files were deposited and available on Gene Expression Omnibus by accession number GSE104506.

## Acknowledgements

We are very grateful to Jay Johnson, Dan Klionsky and Alan Hinnebusch for yeast strains, plasmids and advice. We thank Tom Dever for plasmids and antibodies, Matt Kaeberlein for the Gcn4-luciferase plasmid and Roy Parker for strains. We thank Jonathan Weissman and Nicholas Ingolia for advice on the ribosome profiling method. We thank Cornelia Kurischko for helpful discussions on Ssd1. We thank the Core Facility at the University of Texas MD Anderson Cancer Center for performing the next generation sequencing. This work was supported by NIH grants CA95641 and AG050660 to JKT. KC is supported in part by grants from NIH/NCI (CA208257 and CA207109), DOD (PC160751), and CPRIT (RP150611).

## Additional information

### Competing interests

Jessica K Tyler: Senior editor, *eLife*. The other authors declare that no competing interests exist.

### Funding

| Funder | Grant reference number | Author |
| --- | --- | --- |
| NIH Office of the Director | CA95641 | Jessica K Tyler |
| NIH Office of the Director | AG050660 | Jessica K Tyler |
| NIH Office of the Director | CA208257 | Kaifu Chen |
| NIH Office of the Director | CA207109 | Kaifu Chen |

The funders had no role in study design, data collection and interpretation, or the decision to submit the work for publication.

### Author contributions

Zheng Hu, Formal analysis, Investigation, Writing—original draft; Bo Xia, Validation; Spike DL Postnikoff, Alin S Tomoiaga, Troy A Harkness, Ja Hwan Seol, Investigation; Zih-Jie Shen, Investigation, Writing—review and editing; Wei Li, Project administration; Kaifu Chen, Data curation, Formal analysis, Supervision; Jessica K Tyler, Conceptualization, Supervision, Funding acquisition, Project administration, Writing—review and editing

### Author ORCIDs

Jessica K Tyler http://orcid.org/0000-0001-9765-1659

### Decision letter and Author response

Decision letter https://doi.org/10.7554/eLife.35551.027
Author response https://doi.org/10.7554/eLife.35551.028

## Additional files

### Supplementary files

• Supplemental file 1. Supplemental Table Legends, supplemental table 2 (yeast strains) and supplemental table 3 (plasmids).
DOI: https://doi.org/10.7554/eLife.35551.021

• Supplemental file 2. Supplemental Table 1. Ribo-seq data during aging. The complete list of sorted genes including their gene expression values and translation efficiency calculations.
DOI: https://doi.org/10.7554/eLife.35551.022

• Transparent reporting form
DOI: https://doi.org/10.7554/eLife.35551.023

### Data availability

Next generation sequencing data files were deposited and available on Gene Expression Omnibus by accession number GSE104506.

The following dataset was generated:

| Author(s) | Year | Dataset title | Dataset URL | Database, license, and accessibility information |
|---|---|---|---|---|
| Jessica K Tyler | 2018 | Next generation sequencing data | https://www.ncbi.nlm.nih.gov/geo/query/acc.cgi?acc=GSE104506 | Publicly available at the NCBI Gene Expression Omnibus (accession no. GSE104506) |

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
