## [Decision Letter]

Thank you for submitting your article "Ssd1 and Gcn2 suppress global translational efficiency in replicatively aged yeast, while their activation in young cells extends lifespan" for consideration by *eLife*. Your article has been reviewed by James Manley as the Senior Editor, a Reviewing Editor, and two reviewers. The following individual involved in review of your submission has agreed to reveal his identity: Matt Kaeberlein (Reviewer #2).

The reviewers have discussed the reviews with one another and the Reviewing Editor, Alan Hinnebusch, has drafted this decision to help you prepare a revised submission.

Summary:

This study provides evidence for a dramatic and pervasive reduction in translation in aged yeast cells, which does not result from reductions in mRNA levels and hence can be attributed to decreased translation efficiencies (TEs) of most mRNAs. Ribosome profiling shows that the most abundant and efficiently translated mRNAs, including those encoding ribosomal proteins, exhibit the strongest reductions in TEs. P-bodies also increase with aging, along with expression of Ssd1 and phosphorylated eIF2alpha (eIF2a), but without induction of GCN4 translation; and there appears to be little or no reduction in TOR signaling. Ssd1 overexpression or underexpression from the GAL promoter can extend or reduce replicative lifespan, respectively. The strong reductions in protein synthesis, as judged by reductions in polysomes, are mitigated by deletion of GCN2 (the eIF2a kinase) or SSD1, with an additive effect in the double mutant, indicating independent contributions of both proteins to the inhibition of translation; although protein synthesis is still strongly reduced in old double mutant cells, indicating that other mechanisms must contribute to the translational repression. The authors attempt to show that activation of GCN2 by overexpressing (OE) the gene for tRNAi, presumably by elevating uncharged tRNAi levels, reduces bulk protein synthesis and increases lifespan. This would support the notion that activation of Gcn2 and attendant reduction in protein synthesis in aged cells would contribute to increased lifespan; however, neither response of tRNAi OE was shown to be dependent on Gcn2. Nor did they show that an activated allele of GCN2 would increase lifespan nor (even more importantly) that deletion of GCN2 would reduce lifespan in the manner claimed for reduced Ssd1 expression. Finally, they show that OE of GCN4 increases lifespan in a manner at least partially dependent on the autophagy gene ATG1, without affecting protein synthesis or eIF2a phosphorylation. However, whether deleting GCN4 would reduce lifespan was not addressed. Moreover, considering that GCN4 expression is not induced in old cells, it appears that the effect of GCN4 OE in increasing lifespan might not be a physiologically relevant mechanism in aged cells.

The evidence for a massive reduction in protein synthesis during aging in yeast is very strong and impressive, and it seems clear that Gcn2 and Ssd1 both contribute to this down-regulation, and that Ssd1 is an important regulator of replicative lifespan. However, it is not demonstrated convincingly that Gcn2, nor its target Gcn4, are also important for wild-type replicative lifespan, even though overexpression of Gcn4 can extend lifespan, and this was shown to involve Atg1.

Essential revisions:

It is necessary to modify the writing not to overstate the importance of Gcn2 and Ssd1, and the evidence ruling out TOR inhibition, in the down-regulation of translation in aged cells.

While the authors have evidence that depletion of SSD1 decreases lifespan; it is important to show that deletion of GCN2 also decreases lifespan.

They need to provide evidence that the effects of overexpressing tRNAi on protein synthesis and lifespan require GCN2. Alternatively, they could show that a constitutively activated form of GCN2 would increase lifespan.

They need to more carefully interpret the significance of the finding that GCN4 overexpression increases lifespan, considering that the increased eIF2a phosphorylation observed in aged cells does not evoke increased translation of GCN4, nor does there seem to be evidence that deletion of GCN4 reduces lifespan.

The authors should provide evidence that autophagy is actually being induced on -Gcn4 overexpression, and it would increase confidence in their interpretation if they could show that a second autophagy factor is required for the increased lifespan on Gcn4 overexpression. A more cautious interpretation of the requirement for Atg1 in the increased lifespan evoked by Gcn4 overexpression is required.

*Reviewer #1:*

This study provides evidence for a dramatic and pervasive reduction in translation in aged yeast cells, which does not result from reductions in mRNA levels and hence can be attributed to decreased translation efficiencies (TEs) of most mRNAs. Ribosome profiling shows that the most abundant and efficiently translated mRNAs, including those encoding ribosomal proteins, tend to exhibit the strongest reductions in TEs. P-bodies also increase with aging, along with expression of Ssd1 and phosphorylated eIF2alpha (eIF2a below), but without induction of GCN4 translation; and there appears to be little or no reduction in TOR signaling. Ssd1 overexpression or underexpression from the GAL promoter can extend or reduce replicative lifespan, respectively. The strong reductions in protein synthesis, as judged by reductions in polysomes, are mitigated to some extent, by deletion of GCN2 (the eIF2a kinase) or SSD1, with an additive effect in the double mutant, indicating independent contributions of both proteins to the inhibition of translation; although protein synthesis is still strongly reduced in old double mutant cells, indicating that other mechanisms contribute to the translational repression. They attempt to show that activation of GCN2 by overexpressing the gene for tRNAi, presumably by elevating uncharged tRNAi levels, reduces bulk protein synthesis and increases lifespan; which would support the notion that activation of Gcn2 and attendant reduction in protein synthesis in aged cells would contribute to increased lifespan; however, neither response of tRNAi OE was shown to be dependent on Gcn2. Nor did they show that an activated allele of GCN2 would increase lifespan nor (even more importantly) that deletion of GCN2 would reduce lifespan in the manner claimed for reduced Ssd1 expression. Finally, they show that OE of GCN4 increases lifespan in a manner largely dependent on the autophagy gene ATG1, without affecting protein synthesis or eIF2a phosphorylation. However, they didn't show that deleting GCN4 would reduce lifespan. Moreover, considering that GCN4 expression is not induced in old cells, it appears that the effect of GCN4 OE in increasing lifespan is not a physiologically relevant mechanism in aged cells.

The evidence for a massive reduction in protein synthesis during aging in yeast is very strong and impressive, and it seems clear that Gcn2 and Ssd1 both contribute to this down-regulation. However, I feel that the Abstract gives the mistaken impression that these proteins are the central players in the down-regulation, whereas the data in Figure 3F indicate clearly that other factors also make critical contributions, as the polysome to monosome ratio in the ssd1gcn2 double mutant old cells is still quite low compared to young cells. In addition, it is an overstatement to say that they have ruled out a role for TOR inhibition in the repression based merely on a Western analysis of a single TOR substrate (Rps6) whose phosphorylation is not mechanistically involved in translational activation in yeast. While they have evidence that depletion of SSD1 decreases lifespan (although they would need to confirm the claimed decrease in Ssd1 levels); they have not shown that deletion of GCN2 decreases lifespan. As noted above, they also have not shown that the effects of tRNAi OE on protein synthesis and lifespan require GCN2 and can be attributed to GCN2 activation. In fact, a much better approach would have been to determine whether introducing a constitutively activated form of GCN2 would increase lifespan. Finally, while it is interesting that GCN4 OE can increase lifespan in a manner dependent on ATG1, it does not appear that this is a physiologically relevant effect as their own data shows that the increased eIF2a phosphorylation observed in aged cells does not evoke increased translation of GCN4, nor is there evidence that deletion of GCN4 reduces lifespan. Thus, it seems crucial to show that phosphorylation of eIF2 by GCN2, and its contribution to reducing protein synthesis in aged cells, is actually expanding lifespan, by showing that a deletion of GCN2 reduces lifespan.

There are also many instances in which the paper is not rigorously written, with overstatements of results or interpretations, especially early in the paper where the conclusions seem to rest on results that come later.

– Subsection “Protein synthesis is globally down-regulated in yeast during replicative aging” and Figure 1A: The reduction in polysomes could reflect low mRNA levels rather than reduced translation rates. Bulk polysomes are dominated by ribosomal protein gene (RPG) mRNAs, so repression of these mRNAs would give the same outcome as a reduction in bulk initiation rates. At this point in the paper we haven't yet learned that mRNA levels remain high in old cells. As such, they need to modify the text to address this issue. Also, since mRNAs were not examined in Figure 1A, no statement about mRNA association with monosomes is possible here. Also, the claim that ribosomal subunits are unchanged is not backed up by analysis of free subunit levels using a gradient separation under low Mg2+ concentrations where 80S ribosomes are unstable. At the very least, they need to quantify the gradients to show similar A260 units in the fractions containing ribosomal species.

– Figure 1E and subsection “Overexpression of Ssd1 in young cells represses protein synthesis and extends lifespan”, top. The reduction in polysomes on Ssd1 overexpression (OE) could primarily reflect reduced mRNA levels vs. reduced translation rates. To show reduced initiation rates, they need to probe for specific mRNAs across the gradient and show a shift to smaller polysomes or monosomes, or free mRNP (at the top of the gradient). Also, they haven't confirmed overexpression of Ssd1 by Western analysis. Western analysis is also required to show that the "repressed" Ssd1 level in Figure 1F is actually lower than the native Ssd1 level.

– Figure 2—figure supplement 1. It's not at all obvious that the differences in slopes of these regression lines indicate an increase in intergenic transcription. Either much more explanation is needed, with appropriate references, or independent evidence is required.

– Subsection “Ribo-seq analysis of protein translation during replicative aging”; subsection “Activation of Gcn2, but not TOR inactivation, during replicative aging”: The claim that TEs are reduced the most for the most abundant mRNAs requires substantiation with data analysis. It would be useful to add data showing how the RPG mRNA levels and TEs change (presumably decrease) during aging, compared to all mRNAs.

– subsection “Ribo-seq analysis of protein translation during replicative aging”: the data supporting this claim about there being no reduction in elongation rates must be shown.

– Subsection “Activation of Gcn2, but not TOR inactivation, during replicative aging”: Figure 3—figure supplement 1 is not informative about the group with higher TE in starvation but not in aging. How many such genes are there and what is their GO enrichment? What is the evidence that most are Gcn4-induced? Also, the increase in eIF2a phosphorylation does not necessarily signify Gcn2 activation, as the eIF2a phosphatase could be inhibited or repressed instead (this comment also pertains to subsection “Ssd1 induction and Gcn2 activation both contribute to reduced protein synthesis in old Cells”, and elsewhere in the paper).

– Subsection “Activation of Gcn2, but not TOR inactivation, during replicative aging”: These statements are not substantiated by data presented at this point in the paper, as they have not yet shown that Gcn2 contributes to down-regulation of translation in old cells. Nor do they cite any data regarding the effect of Gcn2 activation on global translation initiation.

– Subsection “Activation of Gcn2, but not TOR inactivation, during replicative aging” and Figure 4: The use of tRNAi for overexpression was not ideal, as this is the initiator tRNA required for all initiation events. While the evidence in panel E supports the idea that tRNAi OE activates Gcn2 and induces GCN4 expression, as noted above, it is important to show that the reduction in polysomes and increase in lifespan on tRNAi OE would be abolished by deleting GCN2; as a reduction in tRNAi charging could affect formation of the key translation initiation intermediate, the eIF2-GTP-Met-tRNAi ternary complex required for the all translation initiation events. In addition, if their interpretation is correct, then expressing a known genetically activated form of Gcn2 should have the same effect on lifespan. Such GCN2c alleles have already been shown to inhibit polysome assembly and it is unclear why tRNAi OE was employed instead, as it could easily have other consequences besides activation of Gcn2.

– Subsection “Overexpression of Gcn4 is sufficient to extend lifespan in an autophagy-dependent Manner” and Figure 5: It's important to confirm using a reporter of a Gcn4 target gene that Gcn4 transcriptional activation is up-regulated in the cells expressing the activated GCN4 allele.

Reviewer #2:

This is a cool study of how replicative aging in yeast impacts bulk translation and mechanisms involved in this process. It builds on a large body of work linking translation to lifespan and provides new mechanistic insights involving Gcn2, Ssd1, and Gcn4. I have only a few comments for the authors to consider.

The autophagy experiment is not as convincing as the authors claim. It looks like overexpression of Gcn4 still extends lifespan in the atg1 mutant, so at best the phenotype is only partially dependent on autophagy. I'm not even sure it's accurate to say that it's "mostly" Atg1-dependent. It would add more confidence for the authors model that Gcn4 overexpression extends lifespan through induction of autophagy if more than a single autophagy factor were tested and if there was evidence provided that autophagy is actually induced. In the absence of additional data, the authors are encouraged to back off on the interpretation for causality.

[Editors' note: further revisions were requested prior to acceptance, as described below.]

Thank you for resubmitting your work entitled "Ssd1 and Gcn2 suppress global translational efficiency in replicatively aged yeast, while their activation in young cells extends lifespan" for further consideration at *eLife*. Your revised article has been reviewed by James Manley (Senior Editor), a Reviewing Editor, Alan Hinnebusch, and two reviewers.

The manuscript has been substantially improved but there are some remaining issues that need to be addressed before acceptance. In particular, there is skepticism that your analysis of translation elongation rates from ribosome profiling data alone is adequate to rule out a general reduction in the rate of translation elongation in aging cells; and you might need to modify or qualify your conclusion on this point. Also, it was felt that your statements that genes with increased translation during aging overlap with those induced on Gcn4 overexpression, and also overlap with genes containing Gcn4 consensus binding sites, are ostensibly at odds with your evidence that Gcn4 translation and Gcn4 target genes found previously to be transcriptionally induced during amino acid starvation are not being induced during aging in the current study. Hopefully, you an clarify these and a few other issues raised by reviewer #1.

*Reviewer #1:*

This paper is much improved by the addition of new experiments and text, which now make it clear that the inhibition of bulk translation via Gcn2 is not contributing significantly to lifespan extension; and that the lifespan extension afforded by genetic activation of Gcn2, with attendant derepression of Gcn4 target genes in the autophagy pathway is an "intervention" in young cells that can extend lifespan but that does not operate normally during aging. There are however some remaining issues to be addressed, including the induction, or lack thereof, of Gcn4 transcriptional target genes during aging; and whether they can rule out reductions in translation elongation as a contributing factor to the massive reduction in protein synthesis that occurs in aged cells-particularly since Gcn2 and Ssd1 together account for only a fraction of the effect.

– Subsection “Ribo-seq analysis of protein translation during replicative aging”: I still feel that more explanation is required in the legend on Figure 2—figure supplement 1 to explain why increased cryptic internal transcription would be expected to increase the slope of this line. Is it because of an increased occurrence of sense transcripts originating from inside the coding sequences that add RNA reads progressively with increasing distance from the TSS? If so, the analysis would have to be excluding antisense internal transcripts, which was not stipulated.

– Subsection “Ribo-seq analysis of protein translation during replicative aging”: Here too, they should explain better what effect a decrease in rate of translation elongation would have on the slope of this line, assuming that the transit time is reduced uniformly at every codon (the simplest model), as the prediction is not at all intuitively obvious. Also, how do they know that the observed difference in slopes is not meaningful? I am very skeptical that this analysis is adequate to rule out a general reduction in the rate of translation, which typical relies on direct measurements of elongation transit times.

Subsection “Increased phosphorylation of the Gcn2 kinase target eIF2α and persistent phosphorylation of the TOR kinase target Rps6 during replicative aging”: the message here is quite confusing. The data in Figures 4E and 4F show that GCN4 translation is not induced in aging despite the increased eIF2 phosphorylation shown in panels A-B. This would be consistent with a lack of transcriptional up-regulation of Gcn4 target genes during aging. Subsection “Increased phosphorylation of the Gcn2 kinase target eIF2α and persistent phosphorylation of the TOR kinase target Rps6 during replicative aging describes genes whose translation is up-regulated by amino acid starvation in data from the Weismann lab that are not up-regulated during aging, which are said to be enriched in genes induced by increased GCN4 translation; however, the evidence that these GO categories are enriched during transcriptional induction by GCN4 is not provided. In fact, the category of amino acid biosynthetic genes, one of the most prominent groups of Gcn4 target genes, is missing in this list; which might indicate that these genes are in fact being induced during aging. (They cite a review article for the transcriptional response to amino acid starvation induced by Gcn4 up-regulation. Instead, they might want to consult a microarray analysis of the Gcn4 transcriptome in Natarajan et al., (2001)). Then they go on to state that genes with increased translation during aging overlap with those induced on Gcn4 overexpression, and with Gcn4 consensus binding sites, which is difficult to square with their evidence that Gcn4 translation and Gcn4 target genes induced during starvation are not being induced during aging. These statements seem contradictory. Finally, they fail to cite the evidence indicating that activation of Gcn2 during amino acid starvation confers a large global reduction in translation initiation comparable to that seen in aging cells, which actually might not be the case.

– Subsection “Intertwined functions of Ssd1 and Gcn2 during aging”: This should be modified to state that other factors clearly contribute to the reduction, given that only partial polysome recovery occurred in old cells lacking both Gcn2 and Ssd1.

– Abstract: This may be misleading, as the Gcn4-dependent transcriptional induction of autophagy genes results from increased translation of GCN4 evoked by Gcn2 activation via unchanged tRNA, not from decreased bulk translation. It would be more accurate to replace "reduced translation" with "activated Gcn2".

– Figure 7: To my knowledge, there is little to no evidence that in budding yeast 4E-BPs are down-regulated by TOR as a means of increasing translational efficiencies of mRNAs in the manner demonstrated in mammalian cells. Furthermore, I believe that the main effect of active TOR in promoting translation is the upregulation of mRNAs encoding ribosomal proteins; and that little or no evidence exists that S6 phosphorylation stimulates translation.

Reviewer #2:

The authors have done an excellent job in addressing my concerns. I have no further suggestions and support publication in its current form.

---

## [Author Response]

Summary:This study provides evidence for a dramatic and pervasive reduction in translation in aged yeast cells, which does not result from reductions in mRNA levels and hence can be attributed to decreased translation efficiencies (TEs) of most mRNAs. Ribosome profiling shows that the most abundant and efficiently translated mRNAs, including those encoding ribosomal proteins, exhibit the strongest reductions in TEs. P-bodies also increase with aging, along with expression of Ssd1 and phosphorylated eIF2alpha (eIF2a), but without induction of GCN4 translation; and there appears to be little or no reduction in TOR signaling. Ssd1 overexpression or underexpression from the GAL promoter can extend or reduce replicative lifespan, respectively. The strong reductions in protein synthesis, as judged by reductions in polysomes, are mitigated by deletion of GCN2 (the eIF2a kinase) or SSD1, with an additive effect in the double mutant, indicating independent contributions of both proteins to the inhibition of translation; although protein synthesis is still strongly reduced in old double mutant cells, indicating that other mechanisms must contribute to the translational repression. The authors attempt to show that activation of GCN2 by overexpressing (OE) the gene for tRNAi, presumably by elevating uncharged tRNAi levels, reduces bulk protein synthesis and increases lifespan. This would support the notion that activation of Gcn2 and attendant reduction in protein synthesis in aged cells would contribute to increased lifespan; however, neither response of tRNAi OE was shown to be dependent on Gcn2. Nor did they show that an activated allele of GCN2 would increase lifespan nor (even more importantly) that deletion of GCN2 would reduce lifespan in the manner claimed for reduced Ssd1 expression. Finally, they show that OE of GCN4 increases lifespan in a manner at least partially dependent on the autophagy gene ATG1, without affecting protein synthesis or eIF2a phosphorylation. However, whether deleting GCN4 would reduce lifespan was not addressed. Moreover, considering that GCN4 expression is not induced in old cells, it appears that the effect of GCN4 OE in increasing lifespan might not be a physiologically relevant mechanism in aged cells.The evidence for a massive reduction in protein synthesis during aging in yeast is very strong and impressive, and it seems clear that Gcn2 and Ssd1 both contribute to this down-regulation, and that Ssd1 is an important regulator of replicative lifespan. However, it is not demonstrated convincingly that Gcn2, nor its target Gcn4, are also important for wild-type replicative lifespan, even though overexpression of Gcn4 can extend lifespan, and this was shown to involve Atg1.

As you noted, the first goal for this work was to comprehensively examine what happens to protein synthesis during replicative aging (for the first time in any organism) – which turned out to be a global reduction. Our other goals were to discern the mechanisms that reduce protein synthesis during aging (which turned out to be Ssd1 protein level increases and eIF2a phosphorylation) and to genetically enhance those mechanisms, in order to potentially extend replicative lifespan. We achieved all these goals, in addition to showing that activation of Gcn4, which doesn't normally happen during aging, can be used to extend lifespan and have delineated that this is via the autophagy pathway. The reviewers’ comments and the summary above also bring up another important component of our work that we clearly needed to do a better job communicating: our work shows that not all mutants that increase protein synthesis shorten replicative lifespan. This is apparent in the fact that *gcn2∆* mutants, although they have increased protein synthesis (in old cells), they do not have reduced lifespan. In agreement, our data indicate that the lifespan extension that is achieved by overexpressing a tRNA that activates Gcn2, was dependent on induction of Gcn4 translation, rather than the globally reduced protein synthesis *per se*. As such, the relationship between global protein synthesis and lifespan is more complex than often thought – indeed the second reviewer also brings up further examples of the disconnect between changes in protein synthesis and replicative lifespan. We have modified our text accordingly to try to better explain these concepts.

Essential revisions:It is necessary to modify the writing not to overstate the importance of Gcn2 and Ssd1, and the evidence ruling out TOR inhibition, in the down-regulation of translation in aged cells.

This is a fair comment and as requested, we have more accurately stated the role of Gcn2, Ssd1 and TOR during aging from our results throughout the text.

While the authors have evidence that depletion of SSD1 decreases lifespan; it is important to show that deletion of GCN2 also decreases lifespan.

As requested, we performed replicative lifespan analyses on *GCN2* deletion strains and show that this causes no significant reduction in RLS in our strain background (new Figure 5F), as was expected from the failure to induce Gcn4 in old cells. This result shows that while the low levels of Gcn2 activation that occurs in old cells is sufficient to reduce global protein synthesis (Figure 4G), this reduction in protein synthesis is insufficient to affect longevity (Figure 5F) and is consistent with our message that the important role of enhanced Gcn2 activation in lifespan extension is to induce Gcn4. We have now discussed this in the results and Discussion sections, in the context of other manipulations that reduced protein synthesis but do not influence lifespan.

They need to provide evidence that the effects of overexpressing tRNAi on protein synthesis and lifespan require GCN2. Alternatively, they could show that a constitutively activated form of GCN2 would increase lifespan.

As requested, we now show that deletion of *GCN2* blocks the ability of tRNAi overexpression to extend lifespan (new Figure 5F) indicating that the tRNAi overexpression is functioning through the stress response pathway. Furthermore, we now show that deletion of *GCN4* blocks the ability of tRNAi overexpression to extend lifespan (new Figure 5G). This is an important result as it indicates that it is the Gcn4-mediated transcriptional changes rather than the global changes in protein synthesis directly, that are extending the replicative lifespan.

They need to more carefully interpret the significance of the finding that GCN4 overexpression increases lifespan, considering that the increased eIF2a phosphorylation observed in aged cells does not evoke increased translation of GCN4, nor does there seem to be evidence that deletion of GCN4 reduces lifespan.

All these observations are exactly in line with our argument that the important part of the Gcn2 stress response in extending lifespan is indeed via Gcn4 induction. This is further validated by our new data showing that deletion of *GCN4* blocks the ability of tRNAi overexpression to extend lifespan (new Figure 5G). This indicates that it is the Gcn4-mediated transcriptional changes rather than the global changes in protein synthesis directly, that are extending the replicative lifespan.

The authors should provide evidence that autophagy is actually being induced on -Gcn4 overexpression, and it would increase confidence in their interpretation if they could show that a second autophagy factor is required for the increased lifespan on Gcn4 overexpression. A more cautious interpretation of the requirement for Atg1 in the increased lifespan evoked by Gcn4 overexpression is required.

As requested, we now show that autophagic flux is induced in cells by overexpression of Gcn4 and is obliterated upon deletion of *ATG1* or *ATG8 (n*ew Figure 6D). We have now additionally included analysis of the effect of *ATG8* deletion on lifespan extension due to Gcn4 overexpression, and show it completely blocks the ability of Gcn4 overexpression to extend replicative lifespan (new Figure 6F). Given that the *ATG1* deletion previously had not completely blocked lifespan extension by Gcn4 overexpression, we went back and examined more Gcn4 overexpressing strains with *ATG1* deletion and found that *ATG1* deletion completely blocks the ability of Gcn4 overexpression to extend replicative lifespan (new Figure 6E) and presumably the isolate that we examined previously had gained a suppressor mutant. These data are consistent with the finding that Gcn4 overexpression activates autophagy to extend lifespan.

Reviewer #1:This study provides evidence for a dramatic and pervasive reduction in translation in aged yeast cells, which does not result from reductions in mRNA levels and hence can be attributed to decreased translation efficiencies (TEs) of most mRNAs. Ribosome profiling shows that the most abundant and efficiently translated mRNAs, including those encoding ribosomal proteins, tend to exhibit the strongest reductions in TEs. P-bodies also increase with aging, along with expression of Ssd1 and phosphorylated eIF2alpha (eIF2a below), but without induction of GCN4 translation; and there appears to be little or no reduction in TOR signaling. Ssd1 overexpression or underexpression from the GAL promoter can extend or reduce replicative lifespan, respectively. The strong reductions in protein synthesis, as judged by reductions in polysomes, are mitigated to some extent, by deletion of GCN2 (the eIF2a kinase) or SSD1, with an additive effect in the double mutant, indicating independent contributions of both proteins to the inhibition of translation; although protein synthesis is still strongly reduced in old double mutant cells, indicating that other mechanisms contribute to the translational repression. They attempt to show that activation of GCN2 by overexpressing the gene for tRNAi, presumably by elevating uncharged tRNAi levels, reduces bulk protein synthesis and increases lifespan; which would support the notion that activation of Gcn2 and attendant reduction in protein synthesis in aged cells would contribute to increased lifespan; however, neither response of tRNAi OE was shown to be dependent on Gcn2. Nor did they show that an activated allele of GCN2 would increase lifespan nor (even more importantly) that deletion of GCN2 would reduce lifespan in the manner claimed for reduced Ssd1 expression. Finally, they show that OE of GCN4 increases lifespan in a manner largely dependent on the autophagy gene ATG1, without affecting protein synthesis or eIF2a phosphorylation. However, they didn't show that deleting GCN4 would reduce lifespan. Moreover, considering that GCN4 expression is not induced in old cells, it appears that the effect of GCN4 OE in increasing lifespan is not a physiologically relevant mechanism in aged cells.The evidence for a massive reduction in protein synthesis during aging in yeast is very strong and impressive, and it seems clear that Gcn2 and Ssd1 both contribute to this down-regulation. However, I feel that the Abstract gives the mistaken impression that these proteins are the central players in the down-regulation, whereas the data in Figure 3F indicate clearly that other factors also make critical contributions, as the polysome to monosome ratio in the ssd1gcn2 double mutant old cells is still quite low compared to young cells. In addition, it is an overstatement to say that they have ruled out a role for TOR inhibition in the repression based merely on a Western analysis of a single TOR substrate (Rps6) whose phosphorylation is not mechanistically involved in translational activation in yeast. While they have evidence that depletion of SSD1 decreases lifespan (although they would need to confirm the claimed decrease in Ssd1 levels); they have not shown that deletion of GCN2 decreases lifespan. As noted above, they also have not shown that the effects of tRNAi OE on protein synthesis and lifespan require GCN2 and can be attributed to GCN2 activation. In fact, a much better approach would have been to determine whether introducing a constitutively activated form of GCN2 would increase lifespan. Finally, while it is interesting that GCN4 OE can increase lifespan in a manner dependent on ATG1, it does not appear that this is a physiologically relevant effect as their own data shows that the increased eIF2a phosphorylation observed in aged cells does not evoke increased translation of GCN4, nor is there evidence that deletion of GCN4 reduces lifespan. Thus, it seems crucial to show that phosphorylation of eIF2 by GCN2, and its contribution to reducing protein synthesis in aged cells, is actually expanding lifespan, by showing that a deletion of GCN2 reduces lifespan.There are also many instances in which the paper is not rigorously written, with overstatements of results or interpretations, especially early in the paper where the conclusions seem to rest on results that come later.

Having found and characterized a global reduction in protein synthesis during aging, we sought to learn the mechanisms that down regulate protein synthesis during aging. To be clear, we had no preconceived requirement that the reduced protein synthesis in old cells had to be regulating longevity. As requested by the reviewer, we have now examined lifespan upon *GCN2* deletion. This new data shows that the Gcn2-mediated down regulation of protein synthesis in old cells has no effect on longevity (new Figure 5F). We thank the reviewer for the suggestion, because this important new result is consistent with the failure to observe Gcn4 activation in old cells (Figure 4C-F), given that a central point of our paper is that the lifespan extension achieved by activation of Gcn2 in young cells is dependent on Gcn4-induced autophagy, rather than global protein synthesis reduction *per se*. Similarly, given that Gcn4 is not activated during normal aging, we had no expectation that *GCN4* deletion would alter normal lifespan. As requested, we performed the lifespan analysis upon *GCN4* deletion and show that as expected, *gcn4∆* has no influence on lifespan (new Figure 5G).

– Subsection “Protein synthesis is globally down-regulated in yeast during replicative aging” and Figure 1A: The reduction in polysomes could reflect low mRNA levels rather than reduced translation rates. Bulk polysomes are dominated by ribosomal protein gene (RPG) mRNAs, so repression of these mRNAs would give the same outcome as a reduction in bulk initiation rates. At this point in the paper we haven't yet learned that mRNA levels remain high in old cells. As such, they need to modify the text to address this issue. Also, since mRNAs were not examined in Figure 1A, no statement about mRNA association with monosomes is possible here. Also, the claim that ribosomal subunits are unchanged is not backed up by analysis of free subunit levels using a gradient separation under low Mg2+ concentrations where 80S ribosomes are unstable. At the very least, they need to quantify the gradients to show similar A260 units in the fractions containing ribosomal species.

We thank the reviewer for this comment, which made us realize that we omitted to mention our previous benchmark RNA-seq analysis during aging that showed that “the abundance of virtually all transcripts encoded by the yeast genome increased during replicative aging (Hu et al., 2014)”, which we have now stated and added at the beginning of the Results section. As requested, we also now specifically show a very subtle change in ribosomal protein gene mRNAs (20% reduction on average for all RPG genes) during aging in new Figure 3C. We have now removed discussions of mRNAs on monosomes and the discussion of ribosomal subunit abundance, in response to the reviewers comment.

– Figure 1E and subsection “Overexpression of Ssd1 in young cells represses protein synthesis and extends lifespan”, top. The reduction in polysomes on Ssd1 overexpression (OE) could primarily reflect reduced mRNA levels vs. reduced translation rates. To show reduced initiation rates, they need to probe for specific mRNAs across the gradient and show a shift to smaller polysomes or monosomes, or free mRNP (at the top of the gradient). Also, they haven't confirmed overexpression of Ssd1 by Western analysis. Western analysis is also required to show that the "repressed" Ssd1 level in Figure 1F is actually lower than the native Ssd1 level.

Overexpression of Ssd1 has not been reported to cause reduced levels of mRNA transcription. Unfortunately, there is not an antibody for endogenous Ssd1, and we did not wish to epitope tag it in case it alters its function. Ssd1 induction and repression was achieved with the very widely used pGAL promoter system. Taken together with the fact that we see clear and opposite phenotypes upon Ssd1 induction and repression, respectively, we are confident that the levels of Ssd1 were induced or repressed as expected by galactose and glucose respectively. Furthermore, high level overexpression of Ssd1 is lethal, and when we increased the galactose concentration to 2% to overexpress Ssd1, we also observed lethality. As such, we are confident that the Gal promoter is altering the expression of Ssd1.

– Figure 2—figure supplement1. It's not at all obvious that the differences in slopes of these regression lines indicate an increase in intergenic transcription. Either much more explanation is needed, with appropriate references, or independent evidence is required.

The changes in slopes in our internal initiation of transcription analysis are significant (indicated by the p values) and are very similar to those in a previous report from Shelly Berger’s group, which we now cite, that only analyzed long genes, as opposed to our analysis that examined all genes. To provide further support for cryptic initiation, we also now examine intergenic transcription during aging and show that also increases during aging (new Figure 2—figure supplement 2).

– Subsection “Ribo-seq analysis of protein translation during replicative aging”; subsection “Activation of Gcn2, but not TOR inactivation, during replicative aging”: The claim that TEs are reduced the most for the most abundant mRNAs requires substantiation with data analysis. It would be useful to add data showing how the RPG mRNA levels and TEs change (presumably decrease) during aging, comparedto all mRNAs.

As requested, we have now added an analysis that shows the translation efficiency is most reduced for the most abundant mRNAs (new Figure 3A, new Figure 3—figure supplement 2). We have also performed the requested analysis of RPG mRNA levels and TE, compared to control mRNAs (new Figure 3B,C).

– Subsection “Ribo-seq analysis of protein translation during replicative aging”: the data supporting this claim about there being no reduction in elongation rates must be shown.

As requested, we have now added an analysis of initiation and elongation during aging (new Figure 2—figure supplement 5).

– Subsection “Activation of Gcn2, but not TOR inactivation, during replicative aging”: Figure 3—figure supplement 1 is not informative about the group with higher TE in starvation but not in aging. How many such genes are there and what is their GO enrichment? What is the evidence that most are Gcn4-induced? Also, the increase in eIF2a phosphorylation does not necessarily signify Gcn2 activation, as the eIF2a phosphatase could be inhibited or repressed instead (this comment also pertains to subsection “Ssd1 induction and Gcn2 activation both contribute to reduced protein synthesis in old Cells”, and elsewhere in the paper).

As requested, we have now added a GO enrichment analysis of the genes whose TE was elevated in starvation but not in aging (new Figure 4D), which indicates that they are in the pathways induced by Gcn4. We have also included an analysis of the genes that are induced by Gcn4 and identification of genes with Gcn4 binding sites (new Figure 4—figure supplement 3). We have altered the text according to the suggestions to discuss levels of eIF2a phosphorylation rather than Gcn2 activity, discussing the possibility of phosphatases being down-regulated. However, we also point out that even if the phosphatases were down-regulated, this would have no consequence in the absence of Gcn2-mediated phosphorylation.

– Subsection “Activation of Gcn2, but not TOR inactivation, during replicative aging”: These statements are not substantiated by data presented at this point in the paper, as they have not yet shown that Gcn2 contributes to down-regulation of translation in old cells. Nor do they cite any data regarding the effect of Gcn2 activation on global translation initiation.

Point taken, and we have adjusted the text accordingly.

– Subsection “Activation of Gcn2, but not TOR inactivation, during replicative aging” and Figure 4: The use of tRNAi for overexpression was not ideal, as this is the initiator tRNA required for all initiation events. While the evidence in panel E supports the idea that tRNAi OE activates Gcn2 and induces GCN4 expression, as noted above, it is important to show that the reduction in polysomes and increase in lifespan on tRNAi OE would be abolished by deleting GCN2; as a reduction in tRNAi charging could affect formation of the key translation initiation intermediate, the eIF2-GTP-Met-tRNAi ternary complex required for the all translation initiation events. In addition, if their interpretation is correct, then expressing a known genetically activated form of Gcn2 should have the same effect on lifespan. Such GCN2c alleles have already been shown to inhibit polysome assembly and it is unclear why tRNAi OE was employed instead, as it could easily have other consequences besides activation of Gcn2.

As requested, we now show that the lifespan extension caused by tRNAi overexpression is dependent on both Gcn2 (new Figure 5F) and Gcn4 (new Figure 5G). So, while tRNAi overexpression affect other events in addition to Gcn2 activation, the functions that are relevant for lifespan extension depend on Gcn2 and Gcn4

– Subsection “Overexpression of Gcn4 is sufficient to extend lifespan in an autophagy-dependent Manner” and Figure 5: It's important to confirm using a reporter of a Gcn4 target gene that Gcn4 transcriptional activation is up-regulated in the cells expressing the activated GCN4 allele.

As requested, we confirmed the expression of Gcn4 using antibodies to endogenous Gcn4, as shown in the new inset in Figure 6A.

Reviewer #2:This is a cool study of how replicative aging in yeast impacts bulk translation and mechanisms involved in this process. It builds on a large body of work linking translation to lifespan and provides new mechanistic insights involving Gcn2, Ssd1, and Gcn4. I have only a few comments for the authors to consider.

*The autophagy experiment is not as convincing as the authors claim. It looks like overexpression of Gcn4 still extends lifespan in the atg1 mutant, so at best the phenotype is only partially dependent on autophagy. I'm not even sure it's accurate to say that it's "mostly" Atg1-dependent. It would add more confidence for the authors model that Gcn4 overexpression extends lifespan through induction of autophagy if more than a single autophagy factor were tested and if there was evidence provided that autophagy is actually induced. In the absence of additional data, the authors are encouraged to back off on the interpretation for causality.*

[Editors' note: further revisions were requested prior to acceptance, as described below.]

The manuscript has been substantially improved but there are some remaining issues that need to be addressed before acceptance. In particular, there is skepticism that your analysis of translation elongation rates from ribosome profiling data alone is adequate to rule out a general reduction in the rate of translation elongation in aging cells; and you might need to modify or qualify your conclusion on this point. Also, it was felt that your statements that genes with increased translation during aging overlap with those induced on Gcn4 overexpression, and also overlap with genes containing Gcn4 consensus binding sites, are ostensibly at odds with your evidence that Gcn4 translation and Gcn4 target genes found previously to be transcriptionally induced during amino acid starvation are not being induced during aging in the current study. Hopefully, you an clarify these and a few other issues raised by reviewer #1.Reviewer #1:This paper is much improved by the addition of new experiments and text, which now make it clear that the inhibition of bulk translation via Gcn2 is not contributing significantly to lifespan extension; and that the lifespan extension afforded by genetic activation of Gcn2, with attendant derepression of Gcn4 target genes in the autophagy pathway is an "intervention" in young cells that can extend lifespan but that does not operate normally during aging. There are however some remaining issues to be addressed, including the induction, or lack thereof, of Gcn4 transcriptional target genes during aging; and whether they can rule out reductions in translation elongation as a contributing factor to the massive reduction in protein synthesis that occurs in aged cells-particularly since Gcn2 and Ssd1 together account for only a fraction of the effect.– Subsection “Ribo-seq analysis of protein translation during replicative aging”: I still feel that more explanation is required in the legend on Figure 2—figure supplement 1 to explain why increased cryptic internal transcription would be expected to increase the slope of this line. Is it because of an increased occurrence of sense transcripts originating from inside the coding sequences that add RNA reads progressively with increasing distance from the TSS? If so, the analysis would have to be excluding antisense internal transcripts, which was not stipulated.

The reviewers understanding of Figure 2—figure supplement 1 is correct, and we apologize for not pointing out that antisense transcripts were removed. We have now noted this in the figure legend.

– Subsection “Ribo-seq analysis of protein translation during replicative aging”: Here too, they should explain better what effect a decrease in rate of translation elongation would have on the slope of this line, assuming that the transit time is reduced uniformly at every codon (the simplest model), as the prediction is not at all intuitively obvious. Also, how do they know that the observed difference in slopes is not meaningful? I am very skeptical that this analysis is adequate to rule out a general reduction in the rate of translation, which typical relies on direct measurements of elongation transit times.

We modified the results to better explain what a defect in elongation would look like in riboseq: “We found no evidence for increased translational pausing or elongation defects during aging which are characterized by increased peaks of ribosomes within ORFs and general enrichment of ribosomes within the ORF relative to the start of the ORF [30], respectively (Figure 2—figure supplement 5) suggesting that the rate-limiting step in protein synthesis during aging is initiation.” We have also added the following statement to the discussion: “Without rigorous and direct measurements of elongation transit times during aging, it is feasible that alterations in elongation may contribute to the reduction in protein synthesis that occurs during aging.”

– Subsection “Increased phosphorylation of the Gcn2 kinase target eIF2α and persistent phosphorylation of the TOR kinase target Rps6 during replicative aging”: the message here is quite confusing. The data in Figure 4E and 4F show that GCN4 translation is not induced in aging despite the increased eIF2 phosphorylation shown in panels A-B. This would be consistent with a lack of transcriptional up-regulation of Gcn4 target genes during aging. Subsection “Increased phosphorylation of the Gcn2 kinase target eIF2α and persistent phosphorylation of the TOR kinase target Rps6 during replicative aging describes genes whose translation is up-regulated by amino acid starvation in data from the Weismann lab that are not up-regulated during aging, which are said to be enriched in genes induced by increased GCN4 translation; however, the evidence that these GO categories are enriched during transcriptional induction by GCN4 is not provided.

The reviewer is correct, Weissman’s paper did not show that the increase in translational efficiency upon amino acid starvation was dependent on *GCN4* per se, so we have corrected the text by removing GCN4 to say that “these tend to encode proteins involved in the stress response pathways that are induced by amino acid starvation [34]” citing the Weissman lab paper.

In fact, the category of amino acid biosynthetic genes, one of the most prominent groups of Gcn4 target genes, is missing in this list; which might indicate that these genes are in fact being induced during aging. (They cite a review article for the transcriptional response to amino acid starvation induced by Gcn4 up-regulation. Instead, they might want to consult a microarray analysis of the Gcn4 transcriptome in Natarajan et al., (2001)).

The figure shows the GO categories of genes who have translational efficiency increases during amino acid starvation (from the Weismann study) but whose translational efficiency does not increase during aging. The Weissman lab analysis actually did not find an increase in translation efficiency of the amino acid synthesis genes during amino acid starvation. They found that they were highly translated (because there was more RNA), but the translational efficiency was not significantly increased. The Natarajan et al., data set that the reviewer suggested that we look at is no longer available online, and our focus was too look at differences in translation efficiency during aging and amino acid starvation. To examine the dependence on Gcn4, our bioinformatics analysis (described in Figure 4—figure supplement 3) showed that these genes that have increased translational efficiency during amino acid starvation but not during aging are transcriptionally induced by Gcn4 and / or have Gcn4 binding sites, by analysis of the data set from (Mittal et al., 2017), consistent with our finding no induction of Gcn4 during aging.

Then they go on to state that genes with increased translation during aging overlap with those induced on Gcn4 overexpression, and with Gcn4 consensus binding sites, which is difficult to square with their evidence that Gcn4 translation and Gcn4 target genes induced during starvation are not being induced during aging. These statements seem contradictory.

We really thank the reviewer for pointing this out to us – this was a text mistake on my part. I meant to write “but not” during aging, rather than “and” during aging. This now reads “The genes that had increased translation during amino acid depletion but not during aging were significantly enriched in the 40 genes that are upregulated during Gcn4 overexpression and with the approximately 60 genes with Gcn4 consensus sites in their promoters.”

Finally, they fail to cite the evidence indicating that activation of Gcn2 during amino acid starvation confers a large global reduction in translation initiation comparable to that seen in aging cells, which actually might not be the case.

We have changed this sentence to talk about stresses that phosphorylate eIF2alpha, rather than activation of Gcn2, because as the reviewer states – there is no data published showing that activation of Gcn2 per se (in the absence of amino acid starvation) reduces translational initiation globally.

– Subsection “Intertwined functions of Ssd1 and Gcn2 during aging”: This should be modified to state that other factors clearly contribute to the reduction, given that only partial polysome recovery occurred in old cells lacking both Gcn2 and Ssd1.

As requested, we have now changed the text to “Additional factors also clearly contribute to the reduction in protein synthesis that occurs during aging”.

– Abstract: This may be misleading, as the Gcn4-dependent transcriptional induction of autophagy genes results from increased translation of GCN4 evoked by Gcn2 activation via unchanged tRNA, not from decreased bulk translation. It would be more accurate to replace "reduced translation" with "activated Gcn2".

We have now made the requested change.

– Figure 7: To my knowledge, there is little to no evidence that in budding yeast 4E-BPs are down-regulated by TOR as a means of increasing translational efficiencies of mRNAs in the manner demonstrated in mammalian cells. Furthermore, I believe that the main effect of active TOR in promoting translation is the upregulation of mRNAs encoding ribosomal proteins; and that little or no evidence exists that S6 phosphorylation stimulates translation.

We thank the reviewer for pointing this out to us, and we have now changed the text and Figure 7 accordingly.

Reviewer #2:

The authors have done an excellent job in addressing my concerns. I have no further suggestions and support publication in its current form.

We thank the reviewer for this very positive comment.